# AUDIO LOTTERY: SPEECH RECOGNITION MADE ULTRA-LIGHTWEIGHT, TRANSFERABLE, AND NOISE-ROBUST

**Shaojin Ding[1*], Tianlong Chen[2*], Zhangyang Wang[2]**
[1]Texas A&M University, [2] University of Texas at Austin
shjd@tamu.edu, {tianlong.chen,atlaswang}@utexas.edu

## ABSTRACT

Lightweight speech recognition models have seen explosive demands owing to a growing amount of speech-interactive features on mobile devices. Since designing such systems from scratch is non-trivial, practitioners typically choose to compress large (pre-trained) speech models. Recently, lottery ticket hypothesis reveals the existence of highly sparse subnetworks that can be trained in isolation without sacrificing the performance of the full models. In this paper, we investigate the tantalizing possibility of using lottery ticket hypothesis to discover lightweight speech recognition models, that are (1) robust to various noise existing in speech; (2) transferable to fit the open-world personalization; and 3) compatible with structured sparsity. We conducted extensive experiments on CNN-LSTM, RNN-Transducer, and Transformer models, and verified the existence of highly sparse "winning tickets" that can match the full model performance across those backbones. We obtained winning tickets that have less than **20%** of full model weights on all backbones, while the most lightweight one only keeps **4.4%** weights. Those winning tickets generalize to structured sparsity with no performance loss, and transfer exceptionally from large source datasets to various target datasets. Perhaps most surprisingly, when the training utterances have high background noises, the winning tickets even **substantially outperform the full models**, showing the extra bonus of noise robustness by inducing sparsity. Codes are available at https://github.com/VITA-Group/Audio-Lottery.

## 1 INTRODUCTION

End-to-end automatic speech recognition (ASR) (Wang et al., 2019; Hannun et al., 2014; Graves, 2012; Chorowski et al., 2015; Dong et al., 2018a) has become an indispensable technology in consumer-interactive devices (e.g., smartphones, smart speakers, tablets) over the past few years (He et al., 2019; Cohen, 2008; Schalkwyk et al., 2010). Conventional on-device ASR systems usually require the involvement of servers, i.e., streaming the audio to servers and then streaming the results back to the devices. By contrast, recent studies (He et al., 2019; McGraw et al., 2016; Kim et al., 2019; Sim et al., 2019; Park et al., 2018; Waibel et al., 2003; Arık et al., 2017) have spurred the success of ASR models fully run on devices, which can be advantageous in terms of computational resources, latency, and user data privacy.

Developing on-device ASR models is challenging since the computational resources (e.g., CPU, memory, battery) are typically very limited. A standard design approach to fit ASR model under budget is through applying various neural network compression techniques to the larger ASR models, such as network pruning (Takeda et al., 2017; Shangguan et al., 2019; Gao et al., 2020), knowledge distillation (Li et al., 2018a), and parameter quantization (He et al., 2019; Sainath et al., 2020). However, there always exists a trade-off between computational efficiency and the model performance, and the efficiency improvements are usually at the cost of word error rate (WER). In most prior works, we observed a non-negligible degradation of WER in compressed models.

---

*Equal Contribution. Shaojin Ding is now with Google. This work was finished when he was with Texas A&M University.

A new horizon emerges with the discovery of lottery tickets hypothesis (LTH) (Frankle & Carbin, 2019b). LTH empirically demonstrated the existence of highly sparse matching subnetworks (i.e., *winning tickets*) in full dense networks, that can be independently trained from scratch to match or even surpass the performance of the latter. LTH was widely observed in various models in computer vision (Liu et al., 2019b; Evci et al., 2019; Frankle et al., 2020c; Savarese et al., 2020; Wang et al., 2020a; You et al., 2020; Ma et al., 2021; Girish et al., 2020) and natural language processing (Gale et al., 2019; Yu et al., 2020; Prasanna et al., 2020; Chen et al., 2020b;c). Yet to our best knowledge, it has not been studied nor utilized in the realm of speech processing and recognition .

This paper presents the first investigation on LTH for developing on-device ASR models. Despite the rich literature of LTH in vision and language, a practically useful winning ticket for real-world ASR would demand two unique properties: **transferability**, **noise robustness**.

- As one persistent challenges of ASR, each individual has a different voice and speaking style. Unlike text or images, whose data are much more "standardized", the spoken word varies greatly based on regional dialects, speed, emphasis, even social class and gender. Therefore, scaling up any ASR system has always been a significant obstacle, since the testing utterances may have very different distribution from the training utterances. That has made transferability a crucial demand for ASR in the open world.

- In an ideal world, one would have to speak very clearly, slowly, and in an environment with no background noise, for the sounds being recognized: that unfortunately will not happen in the practice. In the real-world ASR applications, noise robustness is becoming another crucially demanded technological factor since ASR is now expected to work in much more difficult acoustic environments than in the past (Li et al., 2014). For example, the recognition of speech recorded by distant microphones is challenged by acoustic interference such as noise, reverberation and interference speakers (Kinoshita et al., 2020; Haeb-Umbach et al., 2019; Kinoshita et al., 2013). Even in the standard ASR benchmark such as LibriSpeech (Panayotov et al., 2015), there are significant background noise in its "clean" subset (Zen et al., 2019).

More importantly, prior LTH studies mostly use unstructured sparsity during model pruning. However, designing chips that speeds up unstructured sparse networks are much more complex than those for **structured sparsity** (e.g., Block-Sparse GPU Kernels (Gray et al., 2017)). Together with the two unique gaps, they make ASR no less challenging than visual recognition or text understanding, if not more. They account for the prior arts' difficulty to maintain a satisfactory balance between model efficiency and recognition performance; and similarly, they question the applicability of LTH in ASR.

We provide an affirmative, positive answer of LTH in ASR. As the subject of study, we choose most commonly used ASR architectures in both research and products over the past few years: 1) CNN-LSTM with connectionist temporal classification (CTC) (Graves et al., 2006); 2) RNN-Transducer (Graves, 2012); and 3) Convolution-augmented Transformer (Gulati et al., 2020; Burchi & Vielzeuf, 2021). We conducted extensive LTH experiments using these backbones on three popular corpora: TED-LIUM (Rousseau et al., 2012), Common Voice (Ardila et al., 2020), and LibriSpeech (Panayotov et al., 2015). Unlike most of existing LTH studies that only approached to explaining and demonstrating the correctness of LTH theory, in this work, we make the first attempt to apply LTH to real-world use cases. Namely, we investigate three unique properties that were rarely studied in previous LTH research but are key to user-interactive ASR devices, bringing new insights to both LTH theory and lightweight ASR research. Our main contributions are outlined below:

- We for the **first** time reveal the existence of winning tickets in the context of ASR by answering three research questions regarding LTH theory. The most lightweight winning tickets from CNN-LSTM, RNN-Transducer, and Conformer backbones only possess 21.0%, 10.7%, and 8.6% remaining nonzero weights, respectively. We also show that winning tickets significantly outperform other state-of-the-art network pruning and knowledge distillation methods.

- We are the **first** to explore the use of *structured* sparsity (i.e., block sparsity (Narang et al., 2017b)) in LTH, and successfully found highly sparse winning tickets (e.g., 4.4% remaining weights) that have no performance degradation compared to using *unstructured* sparsity.

- Winning tickets have exceptional transferability across different datasets, which are notably better than full models. Also, the winning tickets identified from large source datasets transfer better.

- In the presence of various levels of background noise, the winning tickets consistently achieve significantly better WERs than full models. That indicates stronger noise robustness might be an extra bonus of inducing sparsity, and **a missing gem** by previous LTH works.

## 2 RELATED WORKS

**End-to-End Automatic Speech Recognition.** Previous mainstream ASR systems are mostly based on hidden Markov model (HMM)-Gaussian mixture model (GMM) or HMM-Deep neural network (DNN). These systems can achieve descent performance. However, such system is usually composed of modules (e.g., acoustic model, language model, lexicon) that are needed to be trained separately, which makes it hard to be optimize globally in products (Wang et al., 2019).

End-to-end ASR can directly transcribe an input audio sequence to a token (e.g., grapheme or phoneme) sequence. Current end-to-end ASR frameworks can be generally categorized into three types: CTC-based (Hannun et al., 2014; Amodei et al., 2016; Graves & Jaitly, 2014; Miao et al., 2015; Eyben et al., 2009), RNN-transducer (Graves, 2012; Graves et al., 2013; Rao et al., 2017; Dong et al., 2018b), sequence-to-sequence (Seq2Seq) model (Chorowski et al., 2015; Bahdanau et al., 2016; Chan et al., 2016; Zhang et al., 2017; Chiu et al., 2018; Prabhavalkar et al., 2018), and Transformer model (Gulati et al., 2020; Dong et al., 2018a; Wang et al., 2020b; Baevski et al., 2020).

CTC (Graves et al., 2006) is essentially a loss function, which maximizes the probability of all the paths that correspond to the ground-truth token sequence, with an augmented blank token "-" indicating no output. It avoids the need of segmental alignment/labels in training utterances, which makes tons of speech materials usable without additional annotating effort and thus fully exploits the modeling capacity of DNNs. RNN-transducer is composed of an encoder for the input audio sequence, a prediction network to model the interdependencies in between the output token sequence, and a joint network to align the input and output sequence and produce the prediction. Seq2Seq model usually has an encoder-decoder structure, where the encoder maps the input audio sequence to a hidden representation sequence and the decoder autogressively decodes the output token sequence. An attention mechanism (Chorowski et al., 2015) is trained to learn the alignment between the input and output sequences. Transformer models are similar to Seq2seq models but use multi-head attention (MHA) (Vaswani et al., 2017) layers for encoders and decoders, which has been proven to achieve the state-of-the-art ASR performance.

**Lottery Tickets Hypothesis.** The recently emerged LTH (Frankle & Carbin, 2019b) deviates from the common wisdom of after-training pruning (Han et al., 2015), and demonstrates the existence of highly sparse subnetworks that are independently trainable from scratch, called *winning tickets*. Once trained, they are capable of matching or even surpassing the performance of their full models. Follow-up efforts (Renda et al., 2020; Frankle et al., 2020b) introduce the effective weight rewinding techniques to scale up LTH to large networks on large-scale datasets. LTH draws wide attention from various deep learning fields, and has been studied in image classification (Liu et al., 2019b; Evci et al., 2019; Frankle et al., 2020c; Savarese et al., 2020; Wang et al., 2020a; You et al., 2020; Ma et al., 2021; Chen et al., 2021d; 2022a;b), natural language processing (Gale et al., 2019; Yu et al., 2020; Prasanna et al., 2020; Chen et al., 2020b;c), object detection (Girish et al., 2020), generative adversarial networks (Chen et al., 2021e; Kalibhat et al., 2020; Chen et al., 2021a), graph neural networks (Chen et al., 2021b), reinforcement learning (Yu et al., 2020), and life-long learning (Chen et al., 2021c). Several pioneer works (Mehta, 2019; Morcos et al., 2019; Desai et al., 2019; Chen et al., 2020b;a) also investigate LTH transferability across datasets and downstream tasks.

Yet to the best of our knowledge, LTH in speech models remains untouched – and that would not be a trivial extension for three aspects of reasons. On the *task* level, ASR has unique demands for individual user transferrability and noise robustness, which has been explained previously. On the *model* level, compared to CV models, speech models are mostly based on RNN backbones (Hannun et al., 2014; Chan et al., 2016), which contain recursive computational graphs and are notoriously unstable to train. That makes the pruning of RNN-based models challenging too (Zhang & Stadie, 2019), and off-the-shelf methods developed for pruning CNNs are often found ineffective or even inferior to random pruning, when applied to RNNs. Pruning methods customized for RNNs typically call for special sparse structures or stability regularizations (Narang et al., 2017a;b; Kliegl et al., 2017; Wen et al., 2018; Zhang & Stadie, 2019), and it is hence unclear whether IMP would remain stable and effective for RNN-based models in ASR. On the *data* level, compared to NLP models, the sequence lengths of speech signals are usually significantly larger than word embeddings (e.g., the spectrogram sequence length of a 10-second speech utterance extracted with 10ms shift is 1,000; the number of words in a sentence is usually between 10 and 50), which also inevitably costs higher computational complexity. Therefore, finding sparse subnetworks that can maintain the full model performance for ASR models is practically meaningful yet highly non-trivial.

# 3 PRELIMINARIES AND SETUPS

**Backbone Network.** We investigate three ASR backbone networks that are widely used in both academia and productions: 1) CNN-LSTM (Amodei et al., 2016) model with CTC loss (Graves et al., 2006); 2) RNN-Transducer (Graves, 2012); and 3) Convolution-augmented Transformer with CTC (Conformer) (Gulati et al., 2020). Please see Appendix A.1 for details of the backbones.

**Datasets, Training, and Evaluations.** We conducted experiments on three commonly used ASR corpora: TED-LIUM (Rousseau et al., 2012) (118 hours), Common Voice (Ardila et al., 2020) (582 hours), and LibriSpeech (Panayotov et al., 2015) (960 hours). Note that LibriSpeech has two test sets: test-clean – little noise inutterances; test-other – considerable noise in utterances. We test the LibriSpeech model on the two test sets hereinafter. During training, we set the batch size to 32 and an initial learning rate to $0.0003$, which is annealed down by a factor of $1.1$ at the end of each epoch. All the models were trained for 16 epochs. To evaluate the performance, we consider two measurements:

• *Word Error Rate.* WER is the standard metric measuring the accuracy of ASR models. WER is computed as: $WER = (S + I + D)/N$, where $S$, $I$, $D$, and $N$ denote the number of substitutions, insertions, deletions, and the total number of words, respectively.

• *Number of Parameters.* The number of parameters measures the complexity of a model. In our case, since all subnetworks were pruned from the full models, we use the percentage of *Remaining Weights* as an alternative measurement. We define *Sparsity* as $\text{Sparsity}(\%) = 100\% - \text{Remaining Weights}(\%)$.

**Subnetworks.** For a dense model $f(x; \theta)$, its subnetworks can be derived as $f(x; m \odot \theta)$ with a binary pruning mask $m \in \{0, 1\}^d$, where $\odot$ is the element-wise product and $d$ is the dimension of pruneable model parameters. We use $\mathcal{A}_t^{\mathcal{D}}(f(x; \theta))$ to represent the training algorithm (e.g., Adam (Kingma & Ba, 2017) with grid searched hyperparameters) that trains a network $f(x; \theta)$ on a dataset $\mathcal{D}$ (e.g., LibriSpeech) for $t$ iterations. Let $\theta_0$ be the random initialized network weights.

**Subnetwork Evaluation.** To measure the generalization ability of obtained subnetworks, we define $\mathcal{E}^{\mathcal{D}}(\mathcal{A}_t^{\mathcal{D}}(f(x; \theta)))$ as the evaluation function of model $f$ returned from $\mathcal{A}_t^{\mathcal{D}}$ on the dataset $\mathcal{D}$. Then, we further introduce:

⋆ *Matching subnetworks.* Following the definition in (Frankle et al., 2020a; Chen et al., 2020b;a), a subnetwork $f(x; m \odot \theta)$ is *matching* if it satisfies the following condition that indicates matching subnetworks achieve **no worse performance than** its dense counterpart under the same training algorithm $\mathcal{A}_t^{\mathcal{D}}$ and evaluation metric $\mathcal{E}^{\mathcal{D}}$: $\mathcal{E}^{\mathcal{D}}\left(\mathcal{A}_t^{\mathcal{D}}\left(f\left(x; m \odot \theta\right)\right)\right) \geq \mathcal{E}^{\mathcal{D}}\left(\mathcal{A}_t^{\mathcal{D}}\left(f\left(x; \theta_0\right)\right)\right)$.

⋆ *Winning ticket.* $f(x; m \odot \theta)$ is a *winning ticket* for $\mathcal{A}_t^{\mathcal{D}}$, if it is $(i)$ a matching subnetwork and $(ii)$ $\theta = \theta_0$ for $\mathcal{A}_t^{\mathcal{D}}$.

⋆ *Transferable Winning ticket.* A subnetwork $f(x; m \odot \theta)$ is *transferable* to **target datasets** $\{\mathcal{D}_i\}_{i=1}^N$ if and only if it is a winning ticket for each $\mathcal{A}_{t_i}^{\mathcal{D}_i}$. The subnetwork $f(x; m \odot \theta)$ is derived from the **source dataset** $\mathcal{D}_s \notin \{\mathcal{D}_i\}_{i=1}^N$.

**Pruning Method for Subnetwork Searching.** Iterative weight magnitude pruning (IMP) is the widely used algorithm in previous LTH literature (Frankle & Carbin, 2019a; Frankle et al., 2020a; Chen et al., 2020b). To identify subnetworks $f(x; m \odot \theta)$, IMP performs following three steps: (1) training a unpruned dense network to completion on a dataset $\mathcal{D}$ (i.e., applying $\mathcal{A}_t^{\mathcal{D}}$); (2) eliminating a portion of insignificant weights with the globally smallest magnitudes (Han et al., 2015; Renda et al., 2020) so that the model only has $s_i\%$ of weights remaining (i.e., the sparsity); (3) rewinding model weights to $\theta$ ($\theta = \theta_0$, the original random initialization; or $\theta = \theta_{pre}$, the weights from a pre-trained model) and finetuning the subnetwork to converge by leveraging $\mathcal{A}_t^{\mathcal{D}}$. Note that steps (2) and (3) usually needs to be iteratively repeated for several rounds for finding highly competitive winning tickets. In all experiments, we set $s_i\% = (1 - 0.8^i) \times 100\%$, where $i$ is the number of iterations.

# 4 THE EXISTENCE OF WINNING TICKETS IN SPEECH RECOGNITION

In this section, we explore the existence of winning tickets in the three ASR backbones. Namely, we would like to answer the following research questions from an empirical perspective:

• *RQ1:* Can we find winning tickets $f(x; m_{IMP} \odot \theta)$ for speech recognition model using IMP? How much do they improve model complexity?

Table 1: Performance of three backbones at the *extreme* sparsity or at the *best* performance on LibriSpeech *test-clean* subset. The performance on *test-other* subset has a similar trend (see Appendix A.3). $\#\mathrm{Params}_{\mathrm{full}}$: number of parameters in full model, in which we use Mega ($\times 10^6$) as the unit; $\mathrm{WER}_{\mathrm{full}}$: WER of full models; $\mathrm{WER}_{\mathrm{ext}}$: WER of the winning tickets at extreme sparsity; $\mathrm{WER}_{\mathrm{best}}$: WER of the best performing winning tickets. Remaining Weight (RW) is included as model complexity measurement.

| Backbone | $\#\mathrm{Params}_{\mathrm{full}}$ | $\mathrm{WER}_{\mathrm{full}}$ | $\mathrm{WER}_{\mathrm{ext}}$ | $\mathrm{RW}_{\mathrm{ext}}$ (#Params) | $\mathrm{WER}_{\mathrm{best}}$ | $\mathrm{RW}_{\mathrm{best}}$ (#Params) |
|---|---|---|---|---|---|---|
| CNN-LSTM | 86.62M | 8.02 | 7.98 | 21.0% (18.19M) | 7.13 | 51.2% (44.34M) |
| RNN-Transducer | 132.23M | 5.90 | 5.71 | 10.7% (14.14M) | 5.39 | 41.0% (54.21M) |
| Conformer | 65.84M | 2.55 | 2.49 | 16.8% (11.06M) | 2.26 | 51.2% (33.71M) |

Table 2: Performance of CNN-LSTM backbone (86.62M parameters) at the *extreme* sparsity or at the *best* performance on TED-LIUM, CommonVoice, and LibriSpeech datasets.

| Dataset | $\mathrm{WER}_{\mathrm{full}}$ | $\mathrm{WER}_{\mathrm{ext}}$ | $\mathrm{RW}_{\mathrm{ext}}$ (#Params) | $\mathrm{WER}_{\mathrm{best}}$ | $\mathrm{RW}_{\mathrm{best}}$ (#Params) |
|---|---|---|---|---|---|
| TED-LIUM | 15.93 | 15.70 | 4.4% (3.81M) | 14.04 | 16.8% (14.55M) |
| CommonVoice | 5.57 | 5.41 | 16.8% (14.55M) | 4.17 | 64.0% (55.43M) |
| LibriSpeech (test-clean) | 8.02 | 7.98 | 21.0% (18.19M) | 7.13 | 51.2% (44.34M) |
| LibriSpeech (test-other) | 20.59 | 20.53 | 21.0% (18.19M) | 19.21 | 51.2% (44.34M) |

- *RQ2:* Do winning tickets identified by IMP have less complexities or better performance, compared to random pruning/random tickets and other compression methods?

- *RQ3:* Instead of using randomized weights $\theta_0$ as the initialization of IMP, does it improve the performance of winning tickets if we use weights $\theta_{pre}$ from a pre-trained model?

**RQ1: Does winning tickets exist in speech recognition models?** To answer the questions, we conducted experiments on three backbones and three datasets. For each trial, we first run IMP to extract a binary pruning mask at each sparsity. Then, we generate one subnetwork at each sparsity by applying the corresponding mask to the model and reset the weights to the original random initialization $\theta_0$. Finally, we train each subnetwork and computes their WER on the test set to determine if they are winning tickets. All the training hyperparameters in training a subnetwork are the same as those in training the full model.

As shown in Table 1, winning tickets can be identified on all three backbones. The most lightweight winning tickets on CNN-LSTM, RNN-Transducer, and Conformer have 21.0%, 10.7%, and 16.8% remaining weights, respectively. In addition, we noticed that the RNN-Transducer subnetworks at the extreme sparsity has less percentage of remaining weights than CNN-LSTM and Conformer subnetworks, likely due to this model being more over-parameterized (RNN-Transducer: 132.23M; CNN-LSTM: 86.62M parameters; Conformer: 65.84M). These results show that, for a fixed dataset, the winning tickets extracted from larger models are sparser.

Similarly, winning tickets can also be identified on all three datasets, as shown in Table 2. We also found that the sparsity of a winning ticket is correlated to the size of the dataset. For example, TED-LIUM has a relatively small size (118 hours) compared to CommonVoice (582 hours) and LibriSpeech (960 hours). Accordingly, the remaining weights of TED-LIUM winning ticket are significantly lower than that of CommonVoice and LibriSpeech winning tickets. A possible explanation is models become relatively more overparameterized for smaller training sets, which allows them to be more amenable for sparsification (Li et al., 2020). Similar observations can be found in (Chen et al., 2020b).

From Table 1 and 2, another interesting finding is that the subnetworks with low sparsity (remain most of the weights) always achieve preferable performance than the full model. We also provided a visualization of the outputs from the full model, the most sparse subnetwork, and the best performing subnetwork in Figure 1 (see Appendix A.4 for more examples.). Similarly, we observed larger performance improvement on smaller datasets, possibly also due to the over parameterization issue. Results indicate that LTH is also a potential research direction in improving overall ASR performance.

**RQ2: Does IMP winning tickets have lower complexity or better performance than random pruning/tickets and other compression methods?** As suggested in previous studies (Frankle & Carbin, 2019a; Chen et al., 2020b; 2021e; 2020a), the two key aspects for a winning ticket to achieve the desired performance are: 1) initial weight $\theta$, and 2) mask generated from IMP $m_{IMP}$. In this subsection, we test if this argument hold in ASR winning tickets. To achieve this, we compare the winning tickets against two baseline pruning approaches: random pruning and random tickets. The subnetworks identified with random pruning are initialized with $\theta_0$ but the masks are randomly generated $m_{RAND}$. By contrast, the subnetworks identified with random tickets are randomly

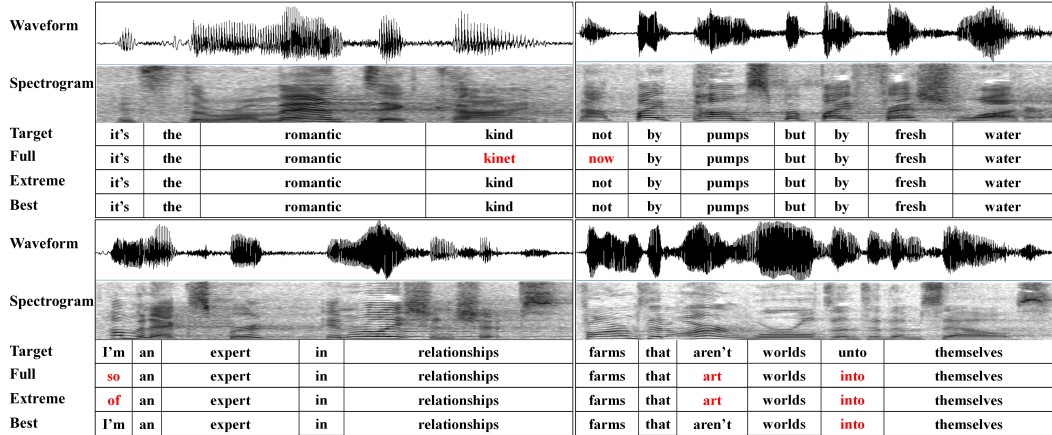

Figure 1: Example outputs from full model, subnetwork at extreme sparsity, and best performing subnetwork on TED-LIUM test utterances. Target: ground-truth transcriptions. Recognition errors are highlighted in red.

initialized $\theta_0'$ but the masks are generated by IMP $m_{IMP}$. As a result, with these two comparisons, we can explore if the two aspects are necessary to a winning ticket.

We tested the three pruning approaches on TED-LIUM dataset. Figure 2 shows the WER curves of the three approaches. From the figure, we found that random pruning and random ticket could find winning tickets, but the extreme remaining weights are much higher than the one identified by IMP (IMP: 4.4%, Random Pruning: 26.2%, Random Ticket: 10.7%). Random pruning can roughly retain its performance when the remaining weights are larger than 26.2%. However, the performance drops dramatically as the remaining weights continue decreasing. Random ticket achieves very similar performance to IMP when the remaining weights are larger than 10.7%, but it degrades much faster than IMP after this point. Results verified the previous statement in ASR models that both initial weight $\theta$ and mask generated from IMP $m_{IMP}$ are necessary for winning tickets. Additionally, the comparison between random pruning and random ticket suggests that $m_{IMP}$ is more important than $\theta$ for winning tickets.

Additionally, we compared our proposed approach against four state-of-the-art neural network compression baselines: 1) Standard Pruning (Han et al., 2015; Blalock et al., 2020; Shangguan et al., 2019); 2) Tutor-Net (Yoon et al., 2021); 3) MLKD + multi-teacher (Li et al., 2021); 4) Sequence-level KD (Takashima et al., 2018). Baseline 1 is the most commonly used and best performing network pruning approach, which iteratively prunes the lowest magnitude weights and train the network until reaching the target sparsity (without any rewinding). Baseline 2, 3, and 4 are

Table 3: Comparison to state-of-the-art pruning and distillation methods on Conformer backbone. Models are evaluated on LibriSpeech *test-clean* subset. See Appendix A.5 for *test-clean* results.

| System | WER | #Params |
|---|---|---|
| Proposed_ext (16.8% RW) | 2.51 | 11.06M |
| Standard Pruning (16.8% RW) | 3.96 | 11.06M |
| TutorNet | 3.86 | 13.09M |
| MLKD + multi-teacher | 13.72 | 11.60M |
| Sequence-level KD | 17.58 | 11.60M |

three recent knowledge distillation approaches. As shown in Table 3, our winning ticket at extreme sparsity (Proposed_ext) achieves superior WER than all the baselines, while using less parameters.

**RQ3: Can IMP find better subnetworks by initializing from a pre-trained model?** Training ASR model from a good initialization usually results in more satisfactory performance (Jaitly et al., 2012), which is commonly achieved by finetuning a pre-trained model in practice. Being aware of this, we would like to investigate if IMP can find better subnetworks for ASR by initializing with weights from a pre-trained model $\theta = \theta_{pre}$. To verify this, we ran IMP on CNN-LSTM backbone with TED-LIUM dataset, where the weights were initialized from either LibriSpeech pre-trained model $\theta_{pre} = \theta_{Libri}$ or CommonVoice pre-trained model $\theta_{pre} = \theta_{CV}$.

WER curves of the subnetworks in this experiment are shown in Figure 3. Both initializing from $\theta_{Libri}$ and $\theta_{CV}$ significantly improves the performance of the subnetworks at any sparsity. More importantly, we found a rapid WER increase when the remaining weights are less than 4.4% for IMP initialized from $\theta_0$. In contrast, the WERs of subnetworks in $\theta_{Libri}$ and $\theta_{CV}$ systems degrades much slower when the remaining weights are extremely low, indicating that the subnetworks that are identified from pre-trained models can utilize the parameters more efficiently. A possible reason is

that the flatter loss surfaces of the pre-trained models (Liu et al., 2019a) make it more traceable for compression, leading to highly quality subnetworks, as also evidenced in (Chen et al., 2020b;a).

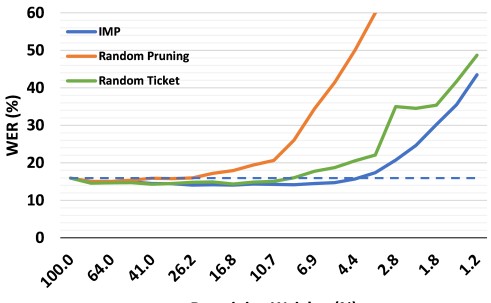

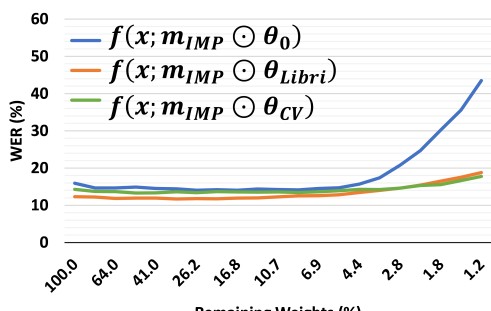

Figure 2: The WER curves of the best subnetworks produced by different pruning approaches. IMP: iterative magnitude pruning, which is the pruning approach we used in subnetwork searching. Random pruning and Random Ticket are the two baseline pruning approaches we evaluated.

Figure 3: The WER curves of initialization with pre-trained models. We test CNN-LSTM backbone on TED-LIUM dataset. $\theta_0$: random initialization; $\theta_{Libri}$: initialized from LibriSpeech pre-trained model; $\theta_{CV}$: initialized from CommonVoice pre-trained model.

Table 4 shows quantitative results of these systems. Initializing from $\theta_{Libri}$ achieves the best performance among all three systems. The WERs of subnetworks initialized from either $\theta_{Libri}$ or $\theta_{CV}$ at different sparsities varies marginally, which makes it hard to find the winning ticket at extreme sparsity. Therefore, we show the WER of all the system with 4.4% remaining weights instead, corresponding to the extreme sparsity of the subnetworks initialized with $\theta_0$. Although WER$_{4.4\%}$ of $\theta_{Libri}$ and $\theta_{CV}$

Table 4: Results of initialization with pre-trained models. WER$_{\text{full}}$: WER of full models; WER$_{\text{best}}$: WER of the best performing matching subnetworks; WER$_{4.4\%}$: WER of subnetworks with 4.4% remaining weights. The brackets shows the remaining weights of the best performing subnetworks.

| Initialization | WER$_{\text{Full}}$ | WER$_{4.4\%}$ | WER$_{\text{Best}}$ |
|---|---|---|---|
| $\theta_0$ | 15.93 | 15.70 | 14.04 (16.8%) |
| $\theta_{Libri}$ | 12.32 | 13.45 | 11.69 (32.8%) |
| $\theta_{CV}$ | 14.30 | 14.28 | 13.3 (51.2%) |

systems are equal or higher than WERs of the corresponding full models, they are still significantly lower than $\theta_0$ system. Additionally, the remaining weights of best performing subnetworks in $\theta_{Libri}$ and $\theta_{CV}$ systems are higher than $\theta_0$, however, when reducing the amount of remaining weights, the WERs only have minimal degradations.

**Summary.** We conducted extensive experiments to answer the three research questions about the existence of winning tickets in three ASR backbones. First, our results verified the existence of winning tickets in CNN-LSTM, RNN-Transducer, and Conformer models, even at high sparsity (e.g., 4.4% remaining weights). Second, we compared IMP with random pruning/tickets and other state-of-the-art network compression methods. The results suggest that matching subnetworks extracted by IMP significantly exceed those extracted by random pruning and random tickets in all measurements, which corroborates both binary pruning mask and weight initialization are indispensable in finding the winning tickets. Our approach also significantly outperforms other compression approaches, achieving state-of-the-art performance on ASR model compression. Lastly, we explored the use of weights from pre-trained models to initialize IMP. We found that, in this way, IMP can identify more effective winning tickets compared to random initialization, and more importantly, these winning tickets have a considerably higher parameter efficiency. These results collectively advocate the profound benefits that LTH can bring to both server-side and on-device ASR.

## 5 Towards Practical ASR with Winning Tickets: Studying Structured Sparsity, Transferablity, and Noise Robustness

After we proved the existence of winning tickets in ASR models, three more properties remain to be verified: structured sparsity, transferability, and noise robustness, which are key to ASR applications.

**Study of structured sparsity** As we mentioned earlier, designing chips that speeds up unstructured sparse networks are much more complex than those for structured sparsity (e.g., Block-Sparse GPU Kernels (Gray et al., 2017)). Therefore, it is critical to verify if we can also identify winning tickets using structured pruning. However, the exploration of structured pruning in prior LTH studies is very limited. (You et al., 2020) is only able to identify limited winning tickets with structured sparsity

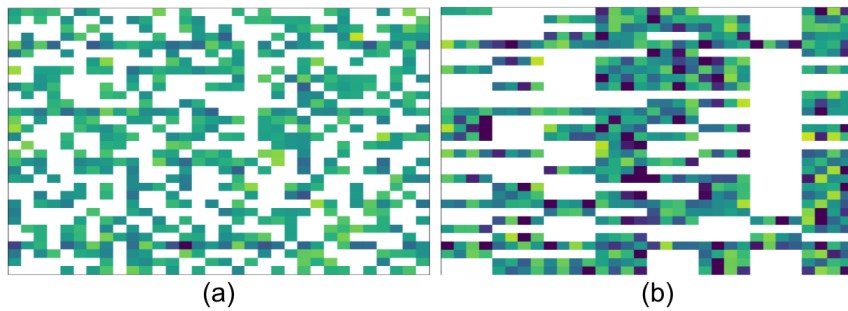

(a)             (b)

Figure 4: Visualizations of weight matrix pruned with (a) unstructured sparsity (b) block sparsity. Pruned weights are shown in white color.

at low sparsity levels (70% remaining weights), which again emphasizes the difficulty in finding computation-friendly sparse patterns.

In this experiment, we applied block sparsity (Narang et al., 2017b; Shangguan et al., 2019; Wu et al., 2021) with 1×4 block to subnetwork searching, and then evaluated the subnetworks to

Table 5: Results of structured sparsity study on TED-LIUM dataset. We also show the results with unstructured sparsity as a reference.

| Sparsity type | $WER_{full}$ | $WER_{ext}$ | $RW_{ext}$ | $WER_{best}$ | $RW_{best}$ |
|---|---|---|---|---|---|
| Unstructured | 15.93 | 15.70 | 4.4% | 14.04 | 16.8% |
| Block sparsity | 15.93 | 15.66 | 4.4% | 13.96 | 21.0% |

see if they can match the full model performance. Results in Table 5 show that using block sparsity during subnetwork searching process lead to comparable results to using unstructured pruning. Also, we visualized the weights of winning ticket models that were discovered with unstructured sparsity and block sparsity, as shown in Figure 4. We can clearly observe 1×4 block patterns from the weight matrix pruned with block sparsity.

**Study of Transferability**   In practical scenarios, the testing utterances are directly recorded from users in the wild, which may have very different distributions from the training utterances. A common way to address this issue is through speaker adaptation (Gauvain & Lee, 1994; Leggetter & Woodland, 1995; Woodland, 2001; Liao, 2013; Li et al., 2018b; Meng et al., 2019; Weninger et al., 2019), which is usually achieved by finetuning a pre-trained model on utterances whose distributions are similar to test utterances directly, or with additional loss terms (Li et al., 2018b; Meng et al., 2019; Ding et al., 2020). In the context of winning tickets, these sparse architectures have to be transferable to new datasets so that we are able to perform speaker adaptation on these models.

To examine the transferability, we conducted the following experiment: First, we identify subnetworks $f(x; m_{IMP} \odot \theta_0)$ on a source dataset $\mathcal{D}_s$ at different sparsities using IMP. Then, we re-train each subnetwork on a target dataset $\mathcal{D}_t$ and evaluate their performance. We tested the transferability between TED-LIUM, CommonVoice, and LibriSpeech datasets. These datasets are different in terms of recording scenario, noise level, speaker coverage, training set size, etc., and therefore, their utterances have very difference distributions. For example, TED-LIUM is created using TED talks, CommonVoice is composed of volunteer's voice recorded through website or mobile apps, and LibriSpeech is extracted from LibriVox audio books.

Figure 5 shows the performance of subnetworks transferring to the three target datasets. Results on the three datasets consistently suggest that the winning tickets are transferable across different datasets. When the remaining weights are larger than around 26.2%, the transferring tickets can generally replicate the performance of the winning tickets identified on the target datasets. However, at extremely high sparsity, the performance of transferring tickets degrades faster than the winning tickets identified on the target datasets, corresponding to the observations from previous studies (Chen et al., 2020c; Morcos et al., 2019; Chen et al., 2021e). For example, Table 9 in Appendix A.6 shows WER and remaining weight of CommonVoice and LibriSpeech winning tickets transferring to TED-LIUM dataset at extreme sparsity. Although the remaining weights of transferring winning tickets are higher than TED-LIUM winning ticket, they are still significantly lower than the full model. We also noticed is that the winning tickets identified from a larger dataset usually have better transferability, which is in accordance with (Morcos et al., 2019).

**Study of Noise Robustness**   The training/adaptation speech utterances are mostly collected from users, which are usually recorded from uncontrolled environments with notable background noise. Even in standard ASR benchmarks such as LibriSpeech (Panayotov et al., 2015), there are significant background noise even in its "clean" subset (Zen et al., 2019). To test the noise robustness of winning tickets, we conducted an experiment on TED-LIUM dataset. Namely, we re-trained the

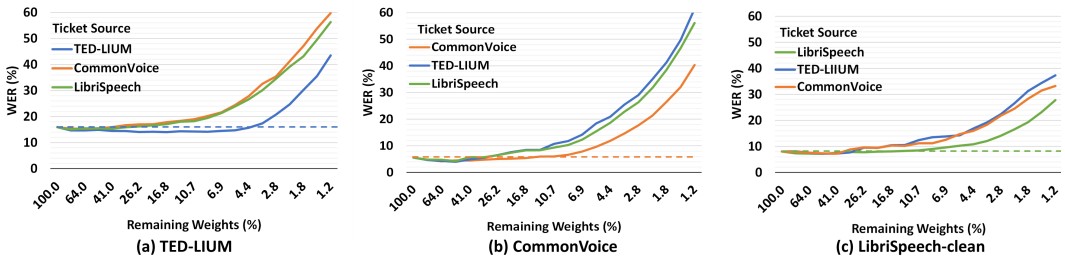

Figure 5: The WER curves of transferring winning tickets to target datasets: (a) TED-LIUM, (b) CommonVoice, and (c) LibriSpeech (test-clean). Each curve represents the winning tickets extracted from a source dataset. The dashed lines indicate the WERs of full models on target datasets.

winning tickets identified from TED-LIUM, CommonVoice, and LibriSpeech on resynthesized TED-LIUM dataset with background noise. We used DESED dataset (Serizel et al., 2020) as the noise source, which consists of various background sounds in domestic environments such as bell, animals, running water, speech, vacuum cleaners, etc. During resynthesis, the noise audio is added to the original speech utterance with a random noise level, which is drawn from a uniform distribution from $[0, N_{max}]$. In addition, we considered three different noise levels: $N_{max} = 0$ (no noise), $N_{max} = 0.2$ (low noise), $N_{max} = 0.5$ (high noise).

Table 6 shows WERs on TED-LIUM test set of the re-trained winning tickets. First, we found the performance of the full model is dramatically susceptible to background noise. When the noise level is increase from $N_{max} = 0$ to $N_{max} = 0.5$, WER increases from 15.93% to 38.21%. In contrast, WERs of TED-LIUM winning ticket at different noise level are very close, indicating that the sparse subnetwork is substaintially more robust to noise. Although the winning tickets from CommonVoice and LibriSpeech cannot reach the WER of TED-LIUM

Table 6: Results of noise robustness study on TED-LIUM dataset. The noise level is drawn from a uniform distribution from $[0, N_{max}]$, and we evaluate three noise conditions: $N_{max} = 0$ (no noise), $N_{max} = 0.2$ (low noise), $N_{max} = 0.5$ (high noise).

| Ticket Source | $N_{max} = 0$ | $N_{max} = 0.2$ | $N_{max} = 0.5$ |
|---|---|---|---|
| Full model | 15.93 | 16.80 | 38.21 |
| TED-LIUM$_{ext}$ | 15.70 | 15.75 | 15.82 |
| CommonVoice$_{ext}$ | 15.93 | 16.73 | 17.53 |
| LibriSpeech$_{ext}$ | 15.88 | 16.23 | 17.38 |
| TED-LIUM$_{best}$ | 14.04 | 14.89 | 14.09 |
| CommonVoice$_{best}$ | 15.32 | 16.27 | 16.79 |
| LibriSpeech$_{best}$ | 15.06 | 16.01 | 16.89 |

winning tickets on noise data, their WERs are still significantly lower than full model. More essentially, we can adapt transfer tickets to any target users without the need of re-finding winning tickets, which is more feasible for practical scenarios. Our findings coincide with the ones in (Ye et al., 2019; Gui et al., 2019) that an appropriate sparsity connectivity severs as implicit regularization for network training, which improves generalization on shifted data distributions (e.g., noisy or perturbed data).

**Summary**   In this section, we examined the structured sparsity, noise robustness, and trasferability of the winning tickets. Throughout the studies, we found that winning tickets can generalize to structured sparsity with no performance degradation. In addition, winning tickets identified from source datasets can achieve matching performance on target datasets, which verifies the transferability. Lastly, winning tickets (identified from either target dataset or source datasets) are significantly more robust to noise compared to full models, especially when the noise level is high. These results jointly demonstrate the eligibility and benefits of winning tickets in on-device ASR applications.

## 6   CONCLUSIONS AND FUTURE WORKS.

In this work, we examine and leverage lottery ticket hypothesis in speech recognition for the first time. Our extensive results show that the winning tickets are not only ultra-lightweight, but also highly transferable and (even more) noise-robust, compared to the full models. These results collectively propound the use of LTH into ASR models, bringing new insights to both LTH theory and portable ASR research. In addition, we also would like to generalize THE LTH compression paradigm to other speech tasks such as text-to-speech synthesis (Wang et al., 2017; Shen et al., 2018) and voice-to-voice conversion (Jia et al., 2019; Biadsy et al., 2019; Zhao et al., 2019). These new directions require further customized studies and point to new opportunities for LTH in speech research.

## ACKNOWLEDGEMENT

Z. Wang is in part supported by the NSF EPCN grant #2053272, and the U.S. Army Research Laboratory Cooperative Research Agreement W911NF17-2-0196 (IOBT REIGN).

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

# A  APPENDIX

## A.1  DETAILS IN BACKBONE ARCHITECTURES

**CNN-LSTM.** The network is composed of two convolutional layers and five recurrent layers, followed by a final fully-connected layer. The first convolutional layer has 32 $41\times11$ (in frequency$\times$time) kernels, with $2\times2$ stride. The second convolutional layer has 32 $21\times11$ kernels, with $2\times1$ stride. For each convlutional layers, *tanh* is used as nonlinearity. Following these, there are five bi-directional RNN layers, each of them having 1,024 units. A final fully-connected layer with softmax operator outputs a probability distribution over characters. The total number of parameters in the full model is 86.62M.

The input to the networks is a sequence of 161-dimensional magnitude spectrogram (160 magnitude points plus 1 DC component), extracted with 20ms window, 10ms shift, and 320-point fast fourier transform. The ground-truth labels are represented by 28 graphemes, including 26 English characters, space, and apostrophe symbol.

**RNN-Transducer.** In RNN-Transducer, we set the encoder and decoder/joint model to have 5 and 1 bidirectional-LSTM layers, each of them having 1,024 units. The total number of parameters in the full model is 132.23M.

The input to the networks is a sequence of 80-dimensional Mel-filterbanks, extracted with 25ms window, 10ms shift, and 512-point fast fourier transform. We used a 1000-dimensional sentence-piece embedding (Kudo & Richardson, 2018) to represent the labels.

**Conformer.** In Conformer, our parameter settings are based on Efficient-Conformer-CTC(L) in (Burchi & Vielzeuf, 2021). Additionally, we reduce the encoder dimensions from $[360, 512, 720]$ to $[256, 384, 512]$, which we found to have no performance degradation and speed up training. The model has 17 encoder layers, each of them is composed of 8-head convolution augmented attention. The convolution kernel has a shape of $1\times32$. The total number of parameters in the full model is 65.84M. During the initial submission, we were using the configurations of Conformer(L) in (Gulati et al., 2020). However, we found that this model is susceptible to gradient explosion when the batch size is limited (e.g., 256). To improve the reproducibility, we switch to Efficient-Conformer-CTC(L) in the camera ready version, which we found to be more stable during training with smaller batch size.

The input to the networks is a sequence of 80-dimensional Mel-filterbanks, extracted with 25ms window, 10ms shift, and 512-point fast fourier transform. We used a 256-dimensional sentence-piece embedding (Kudo & Richardson, 2018) to represent the labels.

## A.2  MORE IMPLEMENTATIONAL DETAILS

As we mentioned in abstract, we will open-source all of the code once the paper is peer-reveiwed. Here, we provide key references and tools that we used during our implementation.

We implement neural network training using PyTorch library (Paszke et al., 2019). Our implementation of CTC backbone is based on:

- https://github.com/SeanNaren/deepspeech.pytorch

Our implementation of RNN-Transduce is based on:

- https://github.com/openspeech-team/openspeech

Our implementation of Conformer is based on:

- https://github.com/burchim/EfficientConformer

We used PyTorch pruning library for unstructured sparsity:

- https://pytorch.org/docs/stable/generated/torch.nn.utils.prune.global_unstructured.html

We implemented pruning functions for structured sparsity based on:

- https://pytorch.org/docs/stable/generated/torch.nn.utils.prune.RandomStructured.html
- https://github.com/openai/blocksparse
- https://www.tensorflow.org/model_optimization/guide/pruning

## A.3 EVALUATING WINNING TICKETS OF THREE ASR BACKBONES ON LIBRISPEECH TEST-OTHER SUBSET

The performance of the three backbones on LibriSpeech *test-other* subset is shown in Table 7.

Table 7: Performance of three backbones at the *extreme* sparsity or at the *best* performance on LibriSpeech *test-other* subset. $\#\text{Params}_{\text{full}}$: number of parameters in full model, in which we use Mega ($\times 10^6$) as the unit; $\text{WER}_{\text{full}}$: WER of full models; $\text{WER}_{\text{ext}}$: WER of the winning tickets at extreme sparsity; $\text{WER}_{\text{best}}$: WER of the best performing winning tickets. Remaining Weight (RW) is included as model complexity measurement.

| Backbone | $\#\text{Params}_{\text{full}}$ | $\text{WER}_{\text{full}}$ | $\text{WER}_{\text{ext}}$ | $\text{RW}_{\text{ext}}$ (#Params) | $\text{WER}_{\text{best}}$ | $\text{RW}_{\text{best}}$ (#Params) |
|---|---|---|---|---|---|---|
| CNN-LSTM | 86.62M | 20.59 | 20.53 | 21.0% (18.19M) | 19.21 | 51.2% (44.34M) |
| RNN-Transducer | 132.23M | 16.93 | 16.30 | 10.7% (14.14M) | 15.55 | 41.0% (54.21M) |
| Conformer | 65.84M | 6.55 | 6.47 | 16.8% (11.06M) | 6.28 | 51.2% (33.71M) |

## A.4 MORE EXAMPLES OF THE MODEL OUTPUTS

We provided a couple more examples of the outputs from full model, winning tickets at extreme sparsity, and best performing winning tickets on TED-LIUM test utterances in Figure 6. The examples in the first row show the cases where the winning tickets avoid the errors appeared in the full model outputs. Meanwhile, we also noticed a few cases where winning tickets produce erroneous recognition results, as shown in the second and the third rows. Most of these utterances have either a very fast speaking rate or unclear pronunciations of words. Therefore, neither full model nor winning tickets reasonably transcribe the utterances in challenging cases, showing the limited capability of the models.

## A.5 COMPARISON TO OTHER MODEL COMPRESSION APPROACHES ON LIBRISPEECH TEST-OTHER SUBSET

Comparison between the proposed approach and state-of-the-art distillation and pruning methods on LibriSpeech *test-other* subset is shown in Table 8. Sequence-level KD approach (Takashima et al., 2018) does not report their evaluation results on test-other subset, and therefore, we use "N/A" in the table.

Table 8: Comparison to state-of-the-art distillation and pruning methods on Conformer backbone. Models are evaluated on LibriSpeech *test-other* subset.

| System | WER | #Params |
|---|---|---|
| Proposed$_{\text{ext}}$ (16.8% RW) | 6.47 | 11.06M |
| Standard Pruning (16.8% RW) | 8.79 | 11.06M |
| TutorNet | 11.14 | 13.09M |
| MLKD + multi-teacher | 34.28 | 11.60M |
| Sequence-level KD | N/A | 11.60M |

## A.6 MORE RESULTS IN TRANSFERRING STUDY

We have shown the WER curves of the transferring winning tickets transferred in Figure 5. Here, we included the WER and RW at the extreme sparsity in Table 9, Table 10 and Table 11, respectively. The transferring tickets to both TED-LIUM and CommonVoice have a extreme remaining weights of 32.8%, and those to LibriSpeech have a extreme remaining weights of 32.8% compared to the full

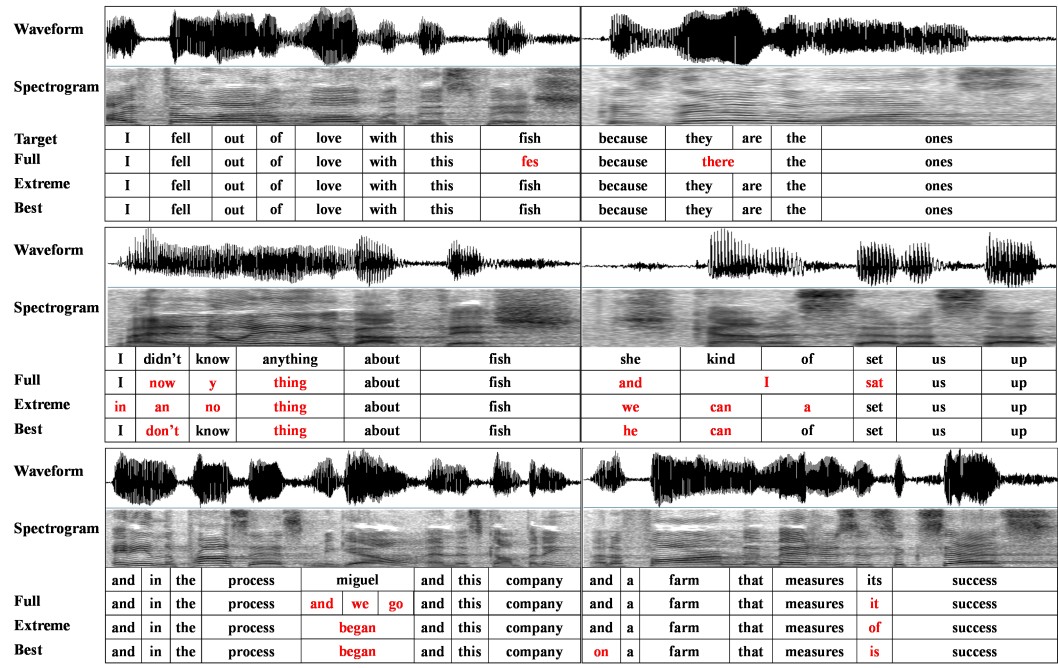

| | | | | | | | | | | | | | | |
|---|---|---|---|---|---|---|---|---|---|---|---|---|---|---|
| **Target** | I | fell | out | of | love | with | this | fish | because | they | are | the | ones | |
| **Full** | I | fell | out | of | love | with | this | fes | because | there | | the | ones | |
| **Extreme** | I | fell | out | of | love | with | this | fish | because | they | are | the | ones | |
| **Best** | I | fell | out | of | love | with | this | fish | because | they | are | the | ones | |

| | | | | | | | | | | | | |
|---|---|---|---|---|---|---|---|---|---|---|---|---|
| | I | didn't | know | anything | about | fish | she | kind | of | set | us | up |
| **Full** | I | now | y | thing | about | fish | and | | I | sat | us | up |
| **Extreme** | in | an | no | thing | about | fish | we | can | a | set | us | up |
| **Best** | I | don't | know | thing | about | fish | he | can | of | set | us | up |

| | | | | | | | | | | | | | | |
|---|---|---|---|---|---|---|---|---|---|---|---|---|---|---|
| | and | in | the | process | miguel | and | this | company | and | a | farm | that | measures | its | success |
| **Full** | and | in | the | process | and we go | and | this | company | and | a | farm | that | measures | it | success |
| **Extreme** | and | in | the | process | began | and | this | company | and | a | farm | that | measures | of | success |
| **Best** | and | in | the | process | began | and | this | company | on | a | farm | that | measures | is | success |

Figure 6: Example outputs from full model, subnetwork at extreme sparsity, and best performing subnetwork on TED-LIUM test utterances. Target: ground-truth transcriptions. Recognition errors are highlighted in red.

models. Although they are not as sparse as the winning tickets identified from the target datasets, the transferring tickets are still much more efficient than full models. These results are in consensus with our findings in Section 5.

Table 9: Performance of transferring CommonVoice and LibriSpeech winning tickets to TED-LIUM dataset at the *extreme* sparsity. Remaining Weight (RW) is included as the spatial and temporal complexity measurements.

| Ticket Source | $WER_{full}$ | WER | RW |
|---|---|---|---|
| TED-LIUM | 15.93 | 15.70 | 4.4% |
| CommonVoice | N/A | 15.93 | 41.0% |
| LibriSpeech | N/A | 15.88 | 32.8% |

Table 10: Performance of transferring TED-LIUM and LibriSpeech winning tickets to CommonVoice dataset at the *extreme* sparsity. Remaining Weight (RW) is included as the spatial and temporal complexity measurements.

| Ticket Source | $WER_{full}$ | WER | RW |
|---|---|---|---|
| CommonVoice | 5.56 | 5.41 | 16.8% |
| TED-LIUM | N/A | 4.25 | 51.2% |
| LibriSpeech | N/A | 5.53 | 32.8% |

Table 11: Performance of transferring TED-LIUM and CommonVoice winning tickets to LibriSpeech dataset at the *extreme* sparsity. Remaining Weight (RW) is included as the spatial and temporal complexity measurements.

| Ticket Source | $\text{WER}_{\text{full}}$ | WER | RW |
|---|---|---|---|
| LibriSpeech | 8.02 | 7.98 | 21.0% |
| TED-LIUM | N/A | 7.72 | 32.8% |
| CommonVoice | N/A | 7.19 | 41.0% |

## A.7 EXTRA NEW RESULTS

### A.7.1 RESULTS ON THE EXISTENCE OF THE THREE BACKBONES ON TED-LIUM AND COMMONVOICE

Table 12: Performance of three backbones at the *extreme* sparsity or at the *best* performance on TED-LIUM. $\#\text{Params}_{\text{full}}$: number of parameters in full model, in which we use Mega ($\times 10^6$) as the unit; $\text{WER}_{\text{full}}$: WER of full models; $\text{WER}_{\text{ext}}$: WER of the winning tickets at extreme sparsity; $\text{WER}_{\text{best}}$: WER of the best performing winning tickets. Remaining Weight (RW) is included as model complexity measurement.

| Backbone | $\#\text{Params}_{\text{full}}$ | $\text{WER}_{\text{full}}$ | $\text{WER}_{\text{ext}}$ | $\text{RW}_{\text{ext}}$ | $\text{WER}_{\text{best}}$ | $\text{RW}_{\text{best}}$ |
|---|---|---|---|---|---|---|
| CNN-LSTM | 86.62M | 15.93 | 15.07 | 4.4% | 14.04 | 16.8% |
| RNN-Transducer | 132.23M | 12.43 | 12.26 | 2.2% | 13.96 | 41.0% |
| Conformer | 65.84M | 7.40 | 7.36 | 3.5% | 7.01 | 21.0% |

Table 13: Performance of three backbones at the *extreme* sparsity or at the *best* performance on CommonVoice. $\#\text{Params}_{\text{full}}$: number of parameters in full model, in which we use Mega ($\times 10^6$) as the unit; $\text{WER}_{\text{full}}$: WER of full models; $\text{WER}_{\text{ext}}$: WER of the winning tickets at extreme sparsity; $\text{WER}_{\text{best}}$: WER of the best performing winning tickets. Remaining Weight (RW) is included as model complexity measurement.

| Backbone | $\#\text{Params}_{\text{full}}$ | $\text{WER}_{\text{full}}$ | $\text{WER}_{\text{ext}}$ | $\text{RW}_{\text{ext}}$ | $\text{WER}_{\text{best}}$ | $\text{RW}_{\text{best}}$ |
|---|---|---|---|---|---|---|
| CNN-LSTM | 86.62M | 5.57 | 5.41 | 16.8% | 4.17 | 64.0% |
| RNN-Transducer | 132.23M | 3.41 | 3.39 | 10.7% | 3.02 | 26.2% |
| Conformer | 65.84M | 1.37 | 1.35 | 8.6% | 1.20 | 20.9% |

### A.7.2 RUN TIME EVALUATIONS

Table 14: Run time evaluation of the three backbones on LibriSpeech at the *extreme* sparsity or at the *best* performance. Here we use the Number of Multiply–Accumulate Operations (MACs) in Giga (G) to measure the run time complexity. We compute the percentage compared to full model for all the subnetworks. $MACs_{full}$: MACs of thefull model. $MACs_{ext}$: MACs of the winning tickets at extreme sparsity. $MACs_{best}$: MACs of the best performing winning tickets.

| Backbone | $MACs_{full}$ | $MACs_{ext}$ | $MACs_{best}$ |
|---|---|---|---|
| CNN-LSTM | 77.88G | 20.1% | 49.9% |
| RNN-Transducer | 124.56G | 9.6% | 39.8% |
| Conformer | 62.27G | 15.9% | 49.4% |

Table 15: Run time evaluation of the three backbones on TEDLIUM at the *extreme* sparsity or at the *best* performance. Here we use the Number of Multiply–Accumulate Operations (MACs) in Giga (G) to measure the run time complexity. We compute the percentage compared to full model for all the subnetworks. $MACs_{full}$: MACs of thefull model. $MACs_{ext}$: MACs of the winning tickets at extreme sparsity. $MACs_{best}$: MACs of the best performing winning tickets.

| Backbone | $MACs_{full}$ | $MACs_{ext}$ | $MACs_{best}$ |
|---|---|---|---|
| CNN-LSTM | 24.34G | 16.0% | 63.1% |
| RNN-Transducer | 38.92G | 1.5% | 39.9% |
| Conformer | 19.45G | 2.9% | 19.7% |

Table 16: Run time evaluation of the three backbones on CommonVoice at the *extreme* sparsity or at the *best* performance. Here we use the Number of Multiply–Accumulate Operations (MACs) in Giga (G) to measure the run time complexity. We compute the percentage compared to full model for all the subnetworks. $MACs_{full}$: MACs of thefull model. $MACs_{ext}$: MACs of the winning tickets at extreme sparsity. $MACs_{best}$: MACs of the best performing winning tickets.

| Backbone | $MACs_{full}$ | $MACs_{ext}$ | $MACs_{best}$ |
|---|---|---|---|
| CNN-LSTM | 53.54G | 20.1% | 49.9% |
| RNN-Transducer | 85.63G | 9.2% | 25.0% |
| Conformer | 42.81G | 7.1% | 19.7% |

### A.7.3 UPDATED NOISE ROBUSTNESS RESULTS

Table 17: Performance of ASR models when adding noise only at test time. Results are shown at the *best* performance.

| Ticket source | no noise | SNR=10dB | SNR=5dB | SNR=3dB | SNR=0dB | SNR=-5dB |
|---|---|---|---|---|---|---|
| Full model | 15.93 | 15.99 | 19.65 | 44.77 | 67.58 | > 100 |
| TEDLIUM$_{best}$ | 14.04 | 14.12 | 16.30 | 20.88 | 40.35 | > 100 |
| CommonVoice$_{best}$ | 15.32 | 15.44 | 17.24 | 21.01 | 45.54 | > 100 |
| LibriSpeech$_{best}$ | 15.06 | 15.25 | 17.49 | 21.50 | 43.98 | > 100 |

### A.7.4 BLOCK SPARSITY EXPERIMENTS WITH LARGER BLOCK SIZE

Table 18: Results of structured sparsity study on TED-LIUM dataset. We also show the results with unstructured sparsity as a reference.

| Sparsity type | WER$_{full}$ | WER$_{ext}$ | RW$_{ext}$ | WER$_{best}$ | RW$_{best}$ |
|---|---|---|---|---|---|
| Unstructured | 15.93 | 15.70 | 4.4% | 14.04 | 16.8% |
| Block sparsity $1 \times 4$ | 15.93 | 15.66 | 4.4% | 13.96 | 21.0% |
| Block sparsity $1 \times 16$ | 15.93 | 15.72 | 4.4% | 14.33 | 16.8% |

### A.7.5 PSEUDO-CODE OF LTH PRUNING ALGORITHM

---
**Algorithm 1** Lottery Ticket Hypothesis Pruning
---
1: Set the initial mask $m$, with the weight initialization $\theta$.
2: **repeat**
3:     Rewind the weight to $\theta$
4:     Train $f(x; m \odot \theta)$ for $t$ epochs with algorithm $\mathcal{A}_t^{\mathcal{P}}$, i.e., $\mathcal{A}_t^{\mathcal{P}}(f(x; m \odot \theta))$
5:     Prune 20% of remaining weights in $\mathcal{A}_t^{\mathcal{P}}(f(x; m \odot \theta))$ and update $m$ accordingly
6: **until** the sparsity of $m$ reaches the desired sparsity level $s$
7: Return $f(x; m \odot \theta)$.
---

### A.7.6 FURTHER EXPLANATION OF BLOCK SPARSITY.

In terms of block sparsity, it prunes a block of parameters (e.g., 1x4 block) instead of just one parameter, which has the minimal magnitude. Unstructured pruning could be thought as block sparsity with 1x1 block.

### A.7.7 EXPLANATION OF THE CHOICE OF SPARSITY LEVELS

In our current setting, we prune out the 20% of remaining weights that have the lowest magnitude. The reason for choosing the value to be 20% is based on considerations of model performance and computation resources. When the amount of weights pruned at each iteration becomes larger, less iterations will be needed. However, it is more likely to have larger performance regressions, since the pruned weights could be useful in the subnetwork at higher sparsity. On the other hand, if the amount of weight pruned at each iteration is too small, the number of iterations required would be very large, which is extremely resource demanding. As a result, we feel 20% is a good one considering both factors.

