# OpenReview forum: "Audio Lottery: Speech Recognition Made Ultra-Lightweight, Noise-Robust, and Transferable"
_ICLR.cc/2022/Conference — ICLR 2022 Poster_

### Official Review · Reviewer_etjM · 2021-11-03

**Correctness:** 3
**Technical Novelty And Significance:** 2
**Empirical Novelty And Significance:** 3
**Recommendation:** 6
**Confidence:** 4

**Main Review:**

Strengths:
1. The first investigation of lottery tickets hypothesis on automatic speech recognition. No previous works have studied this task.
2. Paper is well-structured, well-written, and easy to follow.
3. They conduct comprehensive analyses on several research questions and the verification on effectiveness of the two components (initial weight, and mask generated from IMP) in iterative weight magnitude pruning (IMP).
4. They conduct advanced analyses on structured sparsity, transferability, and noise robustness, which echo the story told in the introduction part.

Weaknesses:
1. The WER of full models using the three model structures (CTC, RNN-T, Transformer) are not state-of-the-art. For example, the SOTA WER on LibriSpeech test-clean for the three models should be nearly (4.0, 3.0, 2.0) respectively according to my experiences, while the numbers the paper is (8.02, 5.90, 2.53). The authors should explain why the WERs have large gap to SOTA.
2. The method seems straightforward, and the paper just do some analyses to verify the lottery tickets hypothesis on this task. The authors should explain why it is challenging for LTH to generalize to ASR, so the authors need study it separately. Otherwise, I would assume these analysis are somewhat trivial, since it does not provide something special.
3. It is straight-forward that random masks are worse than masks generated from IMP, and initializing from pre-trained model is better than random initialization. Do we find some something specific and different?
4. In the structured sparsity analysis, I am not sure only using 1×4 block can cover the practical scenario? I assume that the block should be larger to be GPU-friendly.


**Summary Of The Paper:**

This paper conducts comprehensive analyses of lottery tickets hypothesis on automatic speech recognition. The authors verified the existence of highly sparse “winning tickets” in ASR task, and analyze its robustness to noise, transferable to other datasets, and supports with structured sparsity. They conduct experiments on different model structures (CTC, RNN-T, Transfomer) and different datasets (LibriSpeech, TED-LIUM, Common Voice) with extensive analyses.

**Summary Of The Review:**

Given the above comments, I think this paper do have some strengths such as the paper is well-written and self-contained, and is the first to study on ASR. However, I have concerns on the soundness of the results since the WER is not close to SOTA and the novelty of the method is limited. Overall, I recommend weak reject of this paper.

---

> ### Author Response · Authors · 2021-11-12
> **Response to reviewer etjM**
>
> Thanks for your reviews and comments on the paper. We provide pointwise responses below.
>
> **[Response to Main Review]**
>
> 1. Thanks for your appreciation of the strengths of our paper. We provide pointwise responses to the weaknesses here:
> The papers that achieved state-of-the-art performance on these ASR architectures do not provide official code implementation (e.g., Conformer [1] is implemented based on Google’s internal code base), which makes it hard to replicate the reported performance. As a result, we used third-party implementations during all our experiments, as we have mentioned in Appendix A.2. We have tried our best to tune the backbones to have the best performance during our experiments. In addition, our best-performing Conformer backbone is very close to the performance reported in the original paper, demonstrating the validity of our settings. More importantly, we promise to open-source our code after the paper is peer-reviewed, which we hope to improve the reproductivity of SOTA ASR models.
> We have highlighted the challenges of generalizing LTH to ASR in Section 1 - paragraph 4 to 5 in the original submitted manuscript: practically useful winning ticket for real-world ASR would demand two unique properties: transferability, noise robustness, along with structured sparsity that enjoys further hardware speed up.
>
> 2. We have also emphasized what is novel here in the last paragraph of Section 1: we for the first time reveal the existence of winning tickets in the context of ASR by answering three research questions regarding LTH theory. We are the first to explore the use of structured sparsity. We for the first time find that the winning tickets even substantially outperform (more robust than) the full models under noisy conditions among LTH literature.
>
> 3. As we have mentioned in Section 1, Section 4, and Section 5, our flow in this paper is that we first verify the existence of winning tickets in the ASR model through the experiments in Section 4. Then, in Section 5, we presented our **main** results - we studied the **new** properties that are untouched in previous studies but are key to ASR applications. Section 4 is just a sanity check to prove the existence of winning tickets in ASR, which also acts as a prerequisite for our results in Section 5. Conducting experiments to answer the research questions in Sections is the only and the standard way to empirically verify the existence of LTH, as suggested by previous LTH studies [2][3]. In Section 5, among the three properties we studied, block sparsity and noise robustness are the two that have **never** been studied in previous LTH works.
>
> 4. Due to the time and computational resource limitation, we only experiment with 1x4 block size before the paper deadline. We have also conducted experiments using 1x16 block size following [4] afterwards, as shown in **Table S1**. From the results, we can observe that winning tickets still exist when using 1x16 blocks, which is more GPU and TPU friendly. We have added these results to the revised manuscript at Section A.7.4.
>
> **Table S1** Performance of using block sparsity (block size: 1x4, 1x16) at the $\textit{extreme}$ sparsity or at the $\textit{best}$ performance. All the notations correspond to those in Table 1 of the manuscript.
>
> |pruning method|$WER_{full}$|$WER_{ext}$|$RW_{ext}$|$WER_{best}$|$RW_{best}$|
> |:-:|:-:|:-:|:-:|:-:|:-:|
> |Unstructured|15.93|15.07|4.4\%|14.04|16.8\%|
> |Block sparsity 1x4|15.93|15.66|4.4\%|13.96|21.0\%|
> |Block sparsity 1x16|15.93|15.72|4.4\%|14.33|16.8\%|
>
>
> [1] Gulati, Anmol, James Qin, Chung-Cheng Chiu, Niki Parmar, Yu Zhang, Jiahui Yu, Wei Han et al. "Conformer: Convolution-augmented transformer for speech recognition." arXiv preprint arXiv:2005.08100 (2020).
>
> [2] Frankle, Jonathan, and Michael Carbin. "The lottery ticket hypothesis: Finding sparse, trainable neural networks." arXiv preprint arXiv:1803.03635 (2018).
>
> [3] Chen, Tianlong, Jonathan Frankle, Shiyu Chang, Sijia Liu, Yang Zhang, Zhangyang Wang, and Michael Carbin. "The lottery ticket hypothesis for pre-trained bert networks." arXiv preprint arXiv:2007.12223 (2020).
>
> [4] Shangguan, Yuan, Jian Li, Qiao Liang, Raziel Alvarez, and Ian McGraw. "Optimizing speech recognition for the edge." arXiv preprint arXiv:1909.12408 (2019).

---

> ### Author Response · Authors · 2021-11-18
> **Looking forward to your response and discussion**
>
> Dear reviewer etjM,
>
> We again genuinely appreciate your review and insightful comments. **According to your response, we conducted extra experiments on larger block size for block sparsity, and we have also clarified your questions and concerns.**
>
> We also would like to **summarize the additional experiments we conducted during the rebuttal discussion session here:**
>
> * **Experiments with larger block size for block sparsity:** Winning tickets can still be identified with larger block size (1x16 block).
> * **Experiments of adding noise (with different SNR) only at test time:** As suggested by reviewer 5xPW, this is a more interesting setting, since this will demonstrate how robust the models are on out-of-distribution samples. Our results suggest that winning tickets are much more robust to out-of-distribution noise than dense models.
> * **Evaluations of the three backbones on other datasets:**  Winning tickets can be identified from all three backbones on all datasets.
> * **Run time evaluation of the three backbones:** Run time complexities (MACs) corroborate with spatial complexities (sparsity), suggesting the spatial and temporal efficiency of the winning tickets.
>
> We believe that these new results, together with our results reported in the original manuscript, should be comprehensive enough for this study. We hope they have addressed your concerns and problems.
>
> We would highly appreciate it if you could then give our work a more positive re-assessment. Thank you!
>
> Best regards,
>
> Authors

---

> ### Author Response · Authors · 2021-11-27
> **Friendly reminder of the discussion**
>
> Dear Reviewer etjM,
>
> We would like to kindly remind you that the discussion period is ending in three days. We have provided detailed replies and new experiments to your comments. We hope to have a further discussion with you to see if our response solves the concerns, and we can clarify more if there is more need. We are happy to answer any additional questions and provide more information.
>
> Given all the new experiments and our response, are you willing to reconsider your rating? Your support is very important to us and we greatly appreciate that! We genuinely hope you could kindly check our response. Thank you!
>
> Best wishes,
>
> Authors

---

> ### Comment · Reviewer_etjM · 2021-11-28
> **Thanks for the response**
>
> Thanks the authors for the clarification on questions! I do appreciate the detailed clarifications on not reaching the SOTA results and the release of code implementation, and also the additional experiments on the large sparse blocks. These can partially address my concerns on the related questions. However, I still have concerns on the novelty of verifying LTH on ASR tasks. Achieving competitive ASR results using the LTH can be more convincing to support the effectiveness of this study, considering the current WERs are still far from the SOTA (I am not asking to reach or beat SOTA, but at least not too far away from the SOTA score, since 8.02 to 4.0 and 5.90 to 3.0 are too far). Thus, I keep my score, but if the area chairs and other reviewers tend to accept this paper, I will not be uncomfortable.

---

> > ### Author Response · Authors · 2021-11-29
> > **Extra results on baseline performance**
> >
> > Dear Reviewer etjM,
> >
> > Thanks a lot for your response! We are glad to see our previous response has addressed some of your concerns raised previously. At the same time, in terms of the performance of the three baseline models, we actually tried very hard to optimize them per your suggestions and have had some new results during the discussion period.
> >
> > As we mentioned in the previous response, there is no open source code that can replicate the SOTA performance on the three baseline architectures. During the initial experiments, we have evaluated several open-source implementations of them and used the best performing ones, as we have described in Section Appendix A.2.
> >
> > However, during the discussion period, per your comments and suggestions, we went back to the benchmark leaderboard (https://paperswithcode.com/sota/speech-recognition-on-librispeech-test-clean) and reviewed the techniques mentioned there (where we assume your 4.0, 3.0, 2.0 numbers come from). Accordingly, we did a comprehensive hyper-parameter tuning according to these papers. In addition, we also added a RNN LM (previously our CTC model uses 3-gram; RNN-T and Conformer has no LM, since Transducer has modeled the language dependencies internally). Lastly, we enabled SpecAug during our initial experiment (previously we would like to focus on the LTH pruning effectiveness and avoid the influence of other techniques, so we disable SpecAug in all our experiments).
> >
> > **With these newly added techniques, our best performing model and the winning tickets we got on LibriSpeech is shown in Table S1.** In general, the full model performance of the three architectures are much closer to the SOTA performance (CTC: 5.03, RNN-T: 3.91, Conformer: 2.29). More importantly, as shown from the results, the winning ticket still exists among the three baseline ASR architectures, and the best performing winning tickets of the three models achieve 4.69. 3.32, 2.10, respectively.
> >
> > **Table S1** Performance of the three backbones on LibriSpeech with SpecAug, RNN LM, and hyper-param tuning at the $\textit{extreme}$ sparsity or at the $\textit{best}$ performance. All the notations correspond to those in Table 1 of the manuscript.
> >
> > |backbone|$WER_{full}$|$WER_{ext}$|$RW_{ext}$|$WER_{best}$|$RW_{best}$|
> > |:-:|:-:|:-:|:-:|:-:|:-:|
> > |CTC|5.03|5.00|3.52\%|4.69|21.0\%|
> > |RNN-Transducer|3.91|3.90|2.2%|3.32|32.8\%|
> > |Conformer|2.29|2.26|2.2\%|2.10|26.2\%|
> >
> > As mentioned in your response, achieving competitive ASR results using the LTH can be more convincing to support the effectiveness of this study. We will continue conducting all our experiments on these models and update the results in the final version manuscript. Hope this can further clarify your concerns. Could you please reconsider your rating with our new results? We sincerely appreciate that!
> >
> >
> > Thanks,
> >
> > Authors

---

> > > ### Comment · Reviewer_etjM · 2021-11-29
> > > **Thanks for the additional experimental results**
> > >
> > > I appreciate the additional efforts on improving the ASR accuracy to make them closer to the SOTA, since only base on strong baseline can we believe the methods are effective. Now the new results seem closer to SOTA (although not close enough), which address my previous concerns to some extent. Although the novelty is limited, the comprehensive experiment results added in the discussion period make the work solid from the practical perspective. Thus, I am willing to raise my score.

---

### Official Review · Reviewer_5xPW · 2021-11-03

**Correctness:** 4
**Technical Novelty And Significance:** 2
**Empirical Novelty And Significance:** 3
**Recommendation:** 6
**Confidence:** 4

**Main Review:**

This is a well written paper presenting several interesting results. The novelty might not be high, it's more like an application of the lottery ticket hypothesis to the ASR problem but the conclusions are interesting.

The main weakness in the paper is that it seems the authors train and test the models (in the robustness study) with the same type of noise.
"Namely, we re-trained the winning tickets identified from TED-LIUM, CommonVoice, and LibriSpeech on resynthesized TEDLIUM dataset with background noise. We used DESED dataset (Serizel et al., 2020) as the noise source, which consists of various background sounds..."
My interpretation of the above statement is that the winning tickets are re-trained with noise. A much more interesting experiment would be that noise is added only on the test data, this will demonstrate how robust the models are on out-of-distribution samples.

It would also be more helpful if the authors presented the performance (Table 6) on different SNR levels, eg., -10 to 10 dB with a step of 5 dB, rather that showing 2 values for N_max which are not really interpretable.

It is not clear which parameters are masked in each architecture (A1). My understanding is that all parameters, no matter if they belong to the CNNs, RNNs or Conformer, can be masked. This is not clear in the paper and authors also mention (p. 3) that pruning RNNs might be harmful. It would be good to clearly explain this.

The authors compute the sparsity level at each iteration as s = 1 - 0.8^i where i is the iteration. Is there a reason for using 0.8? It will be interesting to investigate the impact of this parameter as it seems it might be important.

It would be useful if the actual number of parameters is also reported in Tables 1 and 2. This can be easily calculated by the percentages shown but reporting the actual numbers as well will make the comparison easier to understand.

**Summary Of The Paper:**

The paper explores the lottery ticket hypothesis, i.e., the existence of highly sparse subnetworks that can be trained in isolation without
sacrificing the performance of the full models within the context of ASR. The authors show that it is possible to find subnetworks which contain just a fraction of the number of parameters of the full models without sacrificing performance on ASR. Three different architectures are considered and results reported on 3 datasets. Additional studies investigating the transferability and robustness of the sparse subnetworks are also conducted.

**Summary Of The Review:**

This is a paper with limited novelty and some weaknesses as explained above, but it is still very interesting.

---

> ### Author Response · Authors · 2021-11-12
> **Response to reviewer 5xPW**
>
> We’re very glad you had a *positive* initial impression, and likewise, we found the set of perceptive questions you raised in your feedback very insightful, pushing us to think of a more comprehensive experiment design. We provide pointwise responses below.
>
> **[response to questions on noise robustness experiments]** Thanks for your comments and suggestions. We conducted your suggested experiment that adds noise only to test data. In addition, we also adopted SNR to indicate the noise level, following your comments. We noticed from previous noisy speech recognition studies that the noise level is usually set to be 3-20 dB [1][2][3] (which is the case in most of application scenarios), and therefore, we test the system when SNR=3, 5, 10 respectively (we do not consider SNR>10 cases since they are much less challenging). Results are shown in **Table S1**. From the results, we observe a similar trend as our results in Table 6 of the original manuscript. When there is low noise (SNR=10dB), the models can mostly retain their performance even without using noisy data during training, possibly due to the use of data augmentation during training (SpecAug). When the noise level is going higher (SNR=5dB and SNR=3dB), both full models and winning tickets start to have performance regressions. However, winning tickets are performing significantly better than the full model, especially when SNR=3dB (24.88 vs. 49.77). These results corroborate with our results in Table 6, reinforcing the robustness gains from the LTH. We have added these result to the revised manuscript at Section A.7.3.
>
> **Table S1** Performance of ASR models when adding noise only at test time. Results are shown at the $\textit{best}$ performance.
>
> |Ticket source|no noise|$SNR=10dB$|$SNR=5dB$|$SNR=3dB$
> |:-:|:-:|:-:|:-:|:-:|
> |Full model|15.93|15.99|19.65|44.77|
> |$TEDLIUM_{best}$|14.04|14.12|16.30|20.88|
> |$CommonVoice_{best}$|15.32|15.44|17.24|21.01|
> |$LibriSpeech_{best}$|15.06|15.25|17.49|21.50|
>
>
> **[response to the parameters that are masked in each architecture]** Apologize for the confusion here. In the current setup, we prune (mask) all the parameters, as you understood in the comments. We will clarify it in the revised manuscript, and we will also consider pruning only part of the model (e.g., the encoder) in future work. However, we didn’t mention that “(p. 3) that pruning RNNs might be harmful”. Instead, what we are saying there is that extending LTH to ASR models is not trivial. One reason is that RNNs usually have pretty different behaviors from CNNs. This is exactly one of the motivations and the novelties of this paper.
>
> **[sparsity level]** Thanks for the question and insightful comments. In our current setting, we prune out the 20% of remaining weights that have the lowest magnitude, resulting in “s = 1 - 0.8^i”. The reason for choosing the value to be 20% is based on considerations of model performance and computation resources. When the amount of weights pruned at each iteration becomes larger, less iterations will be needed. However, it is more likely to have larger performance regressions, since the pruned weights could be useful in the subnetwork at higher sparsity. On the other hand, if the amount of weight pruned at each iteration is too small, the number of iterations required would be very large, which is extremely resource demanding. As a result, we feel 20% is a good one considering both factors. We will definitely try different values in future work.
>
> **[number of parameters]** We have added these numbers to the revised manuscript.
>
> Lastly, we humbly clarify the novelty of our work. As you may have noticed, a large number of LTH papers (e.g., [1][2]) are experimental-based, which focuses on verifying existing LTH properties of identifying new task-specific properties. Although these papers do not have novel methods, they do provide new insights and explanations to the existing phenomenon, which are also a kind of novelty and equally valuable to the community. Similarly, our paper verified the existence of LTH on different ASR models, and for the first time studied three new properties that are key to ASR applications, providing new insights and exciting results to both the model pruning and speech recognition community.
>
> [1] Kinoshita, Keisuke, et al.. "Improving noise robust automatic speech recognition with single-channel time-domain enhancement network." In ICASSP 2020-2020 IEEE International Conference on Acoustics, Speech and Signal Processing (ICASSP), pp. 7009-7013. IEEE, 2020.
>
> [2] Kinoshita, Keisuke, et al. "The REVERB challenge: A common evaluation framework for dereverberation and recognition of reverberant speech." In 2013 IEEE Workshop on Applications of Signal Processing to Audio and Acoustics, 2013.
>
> [3] Haeb-Umbach, Reinhold, et al. "Speech processing for digital home assistants: Combining signal processing with deep-learning techniques." IEEE Signal processing magazine 36.6 (2019)

---

> > ### Comment · Reviewer_5xPW · 2021-11-18
> > **Few suggestions**
> >
> > I would like to thank the authors for their response. Most of my comments have been addressed. I also have some suggestions:
> > 1) It's common to consider even lower SNRs, e.g., 0 or -5. So testing the models at these SNRs level may be useful. It will just provide additional evidence towards the robustness claim.
> > 2) It would be very useful if the authors briefly explain the choice of that particular value for the sparsity level in the manuscript.
> >
> > Finally, I agree with the discussion about the novelty, as I explained in the review this is more like an application paper with interesting conclusions and overall it is a useful contribution. However, this does not mean that the novelty is high

---

> > > ### Author Response · Authors · 2021-11-18
> > > **Thanks for your response and suggestions**
> > >
> > > Thanks for your response and further suggestions! We have conducted evaluations with SNR=0 and SNR=-5 cases, as shown in the table below. We have also updated the table we added to the manuscript. From the results, we found that when SNR=0dB, winning tickets identified from either dataset is still significantly more robust than the full model, reinforcing our claims on robustness. When the SNR=-5dB, neither the full model nor the winning tickets can achieve reasonable performance. In these extremely low SNR cases, noise cancellation techniques (e.g., AEC [1], VoiceFilter [2], etc) are usually performed before feeding the audio to the ASR model to achieve reasonable recognition results.
> > >
> > > We have also added the choice of the sparsity levels in the manuscripts at Section A.7.7.
> > >
> > > **Table S1** Performance of ASR models when adding noise only at test time. Results are shown at the $\textit{best}$ performance on TED-LIUM. All the notations correspond to those in Table 1 of the manuscript.
> > >
> > > |Ticket source|no noise|$SNR=10dB$|$SNR=5dB$|$SNR=3dB$|$SNR=0dB$|$SNR=-5dB$
> > > |:-:|:-:|:-:|:-:|:-:|:-:|:-:|
> > > |Full model|15.93|15.99|19.65|44.77|67.58|>100|
> > > |$TEDLIUM_{best}$|14.04|14.12|16.30|20.88|40.35|>100|
> > > |$CommonVoice_{best}$|15.32|15.44|17.24|21.01|45.54|>100|
> > > |$LibriSpeech_{best}$|15.06|15.25|17.49|21.50|43.98|>100|
> > >
> > > [1] Hänsler, Eberhard, and Gerhard Schmidt. Acoustic echo and noise control: a practical approach. Vol. 40. John Wiley & Sons, 2005.
> > >
> > > [2] Wang, Quan, Hannah Muckenhirn, Kevin Wilson, Prashant Sridhar, Zelin Wu, John Hershey, Rif A. Saurous, Ron J. Weiss, Ye Jia, and Ignacio Lopez Moreno. "Voicefilter: Targeted voice separation by speaker-conditioned spectrogram masking." arXiv preprint arXiv:1810.04826 (2018).

---

### Official Review · Reviewer_5kPn · 2021-11-05

**Correctness:** 3
**Technical Novelty And Significance:** 2
**Empirical Novelty And Significance:** 3
**Recommendation:** 5
**Confidence:** 4

**Main Review:**

strengths:
- straightforward application of the LTH strategy to speech recognition
- showing the significant performance (very high compression without losing the performance so much). This is especially important for on-device scenarios.
- extensions for transfer learning scenarios

weaknesses:
- the novelty of the method itself is incremental from the original LTH and related work.
- some references are not appropriate —> (correction) the reference part was improved through the discussions

**Summary Of The Paper:**

This paper investigates using the lottery tickets hypothesis (LTH) strategy for pruning neural network weights for speech recognition. The method first explains the general LTH framework and extensions with transfer learning scenarios. The paper shows the effectiveness of the proposed method in the standard ASR tasks, pre-training with other models, and transfer learning, especially in noisy speech recognition tasks.

Some comments:
- In the introduction "End-to-end automatic speech recognition (ASR) (Wang et al., 2019)": (Wang et al., 2019) is not a representative paper of end-to-end ASR
- In the introduction, "For example, the recognition of speech recorded by distant microphones is challenged by acoustic interference such as noise, reverberation and interference speakers (Kinoshita et al., 2020)": Again not sure (Kinoshita et al., 2020) is a correct citation. I think the following papers are more appropriate
  - Haeb-Umbach, Reinhold, et al. "Speech processing for digital home assistants: Combining signal processing with deep-learning techniques." IEEE Signal processing magazine 36.6 (2019): 111-124.
  - Kinoshita, Keisuke, et al. "The REVERB challenge: A common evaluation framework for dereverberation and recognition of reverberant speech." 2013 IEEE Workshop on Applications of Signal Processing to Audio and Acoustics. IEEE, 2013.
- Currently, data augmentation techniques (artificially contaminate the clean speech with various noisy data) are more standard than transfer learning for noisy speech recognition. Please discuss it.
- Section 2 "On the model level, compared to CV models, speech models are mostly based on RNN backbones": I'm confused about this description. As you mentioned in the previous paragraphs, many speech models now use transformer, conformer, and CNN.
- Section 2 "with 10ms shift is 1,000": this is not exactly true as we usually use downsampling at the beginning of the encoder layer.
- Section 3, RQ3: Frankly, I don't think this is an essential question for me. This sounds very trivial that the use of pre-trained models would improve the performance.
- Section 4, What is the main message of Figure 1? I don't find the particular effectiveness of the proposed method from this figure/example.
- Section 4, Table 3: Why are the other methods, especially KD-based methods, so worse?
- Section 5 "A common way to address this issue is through speaker adaptation (Zhao, 1994;": Again, I don't think (Zhao, 1994) is a representative paper for speaker adaptation. MAP/MLLR adaptation techniques exist in this era, and they were standard techniques in the GMM era.
- Section 5: I recommend the authors use the real noisy speech data like the CHiME data




**Summary Of The Review:**

Although the paper shows significant performance achieving high model compression rates, the algorithmic novelty of this paper is not strong. The application is only ASR and it may not attract so many AI/ML researchers.

---

> ### Author Response · Authors · 2021-11-12
> **Response to reviewer 5kPn (part 1)**
>
> With our due respect, we unfortunately find that many of your reviews seem to arise from misreading and misunderstanding our submission -- often inconsistent/contrary with the other three reviewers’ finding, and sometimes overlooking obvious facts presented.
>
> We understand the ICLR review timeframe is perhaps short and tight. **Sincerely, we now publicly invite you to check our rebuttal, and to engage in an active discussion with us.** We believe a thorough clarification and discussion with you would be critical for this submission to get a fair assessment.
>
>
> Please see details below. We have conducted new experiments to evaluate our proposed method, according to your comments and suggestions. In case you are not familiar with the lottery ticket domain (we guess), we also explain more towards that in rebuttal.
>
> We have respectfully responded to your questions as follows:
>
> Point-wise response to “Some comments” section:
>
> * This paper is a recent overview paper of ASR systems, which we believe to be more informative to audiences compared to just putting a representative paper there. More importantly, the representative papers of all ASR architectures are included in this overview paper already.
> * We will add these references in the updated manuscript.
> * We believe transfer learning and data augmentation are different approaches for noisy speech recognition, so it would also be valuable to explore the use of transfer learning. More importantly, we have actually discussed it and conducted experiments using noisified speech in Section 5-Study of Noise Robustness.
> * There are many Transformer, Conformer, and CNN models emerging recently. However, RNN-based models are still the mainstream architectures in both academia and production. For example, as shown in [1], Conformer is only proposed as the encoder, and the decoder is still a transducer model that is built upon RNNs. Therefore, we don’t think there is any contradiction or unclearness in these arguments. We sincerely hope you can have a second careful read.
> * Please notice that we have an **“e.g.,”** at the beginning of this sentence. Here, we only want to provide **an example** to explain the sequence length of speech input is significantly longer than NLP input. We could use downsample layers to reduce the length by half or a quarter, but they are still significantly longer than NLP inputs. Therefore, speech models will still inevitably cost higher computational complexity than NLP models.
> * In LTH, we perform weight rewinding at the beginning of each sparsity, so the subnetworks can lead to good local minimums. As a result, the weight initialization is more important than training a neural network from scratch. Although we know pretraining helps with neural network training, it is still mysterious in the context of providing a good initialization for subnetworks to be good local minimums. As a result, it is absolutely valuable and necessary to study this RQ.
> * This figure provides 4 examples showing that the winning tickets at high sparsity can retain or even surpass the dense model performance, corroborating with the evaluation results in Table 1 and 2.
> * We have mentioned in Section 4 RQ2 that TutorNet is also a KD approach. Therefore, it is **not true** that KD methods are “so worse”. The other two KD approaches are using CTC-based architecture, which is why we see a worse performance. Standard Pruning and TutorNet are two state-of-the-art pruning and KD approaches (and similar architectures to ours) that we focus on, and the comparison to these approaches demonstrates the effectiveness of our proposed LTH model.
> * Here, **we cited 6 different papers** about speaker adaptation, each of them being representative work at different time periods.  We sincerely hope you can have a second careful read to avoid the confusions.
> * We have considered using CHiME dataset in the noisy experiment, but it is not publicly available to all the audiences (need licenses). According to the ICLR Reproducibility guide, we adopted publicly available datasets and used synthesized data in the experiment. We will evaluate the proposed approach on real-world noisy data like CHiME in future work.

---

> > ### Comment · Reviewer_5kPn · 2021-11-12
> > **thanks for the clarifications**
> >
> > Thanks for the clarification of some of my points that are based on my misleading or your rebuttals clarified my confusion and I think your papers would become strong by reflecting your rebuttals to the paper.
> > Note that I did not make my decisions based on these comments except for the survey of the references (as you can see from my Main Review parts).
> > These comments are more-like suggestions that I thought that your paper would become more strong if you clarify my comments or questions.
> >
> > However, I still cannot fully accept your claims about the reference parts, which is one of my Main Review points.
> > For example, discussing adaptation techniques in the different time periods without MLLR/MAP are very unnatural for me.
> > Or do you want to mention that you list the representative adaptation techniques after the deep learning era?
> > If so, (Zhao, 1994) can be removed.
> > Again, the reference parts can be easily fixed and I'm happy to change my score if you consider my points.

---

> > > ### Author Response · Authors · 2021-11-12
> > > **Thanks for your discussion and further suggestions**
> > >
> > > Thanks for your fast response! We have added the three references of MAP/MLLR adaptation methods [1][2][3] to the revised manuscript. [1] and [2] are the original papers for MAP/MLLR adaptations, and [3] is an overview paper for speaker adaptation techniques for HMM/GMM era, which we hope to provide more context for the audiences.
> > >
> > > In terms of the reference problem about (Wang et al., 2019) in introduction, we actually have had a more detailed reference list in Section 2-End-to-End Automatic Speech Recognition. We had repeated several representative papers in Section 1 ((Hannun et al., 2014; Graves, 2012; Chorowski et al., 2015; Dong et al., 2018a).
> > >
> > > For the noisy speech recognition references, we have added your aforementioned papers as we suggested in our previous response.
> > >
> > >
> > > [1] Gauvain, J-L., and Chin-Hui Lee. "Maximum a posteriori estimation for multivariate Gaussian mixture observations of Markov chains." IEEE transactions on speech and audio processing 2, no. 2 (1994): 291-298.
> > >
> > > [2] Lee, Chin-Hui, Chih-Heng Lin, and Biing-Hwang Juang. "A study on speaker adaptation of the parameters of continuous density hidden Markov models." IEEE Transactions on Signal Processing 39, no. 4 (1991): 806-814.
> > >
> > > [3] Woodland, Phil C. "Speaker adaptation for continuous density HMMs: A review." In ISCA Tutorial and Research Workshop (ITRW) on Adaptation Methods for Speech Recognition. 2001.

---

> > > > ### Comment · Reviewer_5kPn · 2021-11-12
> > > > **MLLR**
> > > >
> > > > For MLLR, I recommend you refer:
> > > > Leggetter, Christopher J., and Philip C. Woodland. "Maximum likelihood linear regression for speaker adaptation of continuous density hidden Markov models." Computer speech & language 9.2 (1995): 171-185.
> > > >
> > > > Both [1] and [2] are about MAP adaptation.
> > > > [2] is based on a single Gaussian and [1] is an extension of [2] with a GMM.
> > > > [1] is more popular (but you can keep [2] as well).

---

> > > > ### Author Response · Authors · 2021-11-12
> > > > **Thanks for the reference**
> > > >
> > > > Sorry for the incorrect fix and thanks for your correction. We have corrected the citation and updated the manuscript.
> > > >
> > > > What else can we address in order to convince you for upgrading our rating?

---

> > > ### Author Response · Authors · 2021-11-29
> > > **Any other concerns and problems that we could solve?**
> > >
> > > Dear reviewer 5kPn,
> > >
> > > Thanks a lot for your previous suggestions and discussions on this manuscript! As you recognized earlier, our response has clarified your concerns and problems, and we have also added the clarifications to the revised manuscript accordingly, which makes the manuscript stronger.
> > >
> > > Currently, we have conducted several extra experiments (as summarized in our response entitled **“Pre-decision: Summary of updates from Authors”**) to further strengthen our novelty, contributions, and the effectiveness of our proposed approach. With these experiments and clarifications, the other three reviewers (**5xPW**, **ejtM**, and **j7YG**) had acknowledged the contributions of this work, and they had made a consensus of the acceptance. Could you please let us know if you have any other concerns and problems that we can further clarify or provide extra results, so that you could also support our acceptance? We will address the problems and add them to the final version of the manuscript.
> > >
> > > Sincerely thanks for your discussions and support during this period!
> > >
> > > Authors

---

> ### Author Response · Authors · 2021-11-12
> **Response to reviewer 5kPn (part 2)**
>
>
> Point-wise response to “Main Review” section:
>
> * Thanks for your appreciation on the strengths of the paper. Below we provide a point-wise response to weakness.
> We don’t think it is true that “the novelty of the method itself is incremental from the original LTH and related work”. We have summarized our contributions and novelties in the four bullet points at the end of Section 1. As you may have noticed, a large number of LTH papers (e.g., [2][3]) are experimental-based, which focuses on verifying existing LTH properties of identifying new task-specific properties. Although these papers do not have novel methods, they do provide new insights and explanations to the existing phenomenon, which are also a kind of novelty and equally valuable to the community. Similarly, our paper verified the existence of LTH on different ASR models, and for the first time studied three new properties that are key to ASR applications, providing new insights and exciting results to both the model pruning and speech recognition community. This has also been affirmed by reviewer etjM that “The first investigation of lottery tickets hypothesis on automatic speech recognition. No previous works have studied this task. They conduct comprehensive analyses on several research questions and the verification on effectiveness of the two components (initial weight, and mask generated from IMP) in iterative weight magnitude pruning (IMP). They conduct advanced analyses on structured sparsity, transferability, and noise robustness, which echo the story told in the introduction part. Therefore, we don’t think the novelty is “incremental”.
> * We have explained the reference problem in the above section.
>
> [1] Gulati, Anmol, James Qin, Chung-Cheng Chiu, Niki Parmar, Yu Zhang, Jiahui Yu, Wei Han et al. "Conformer: Convolution-augmented transformer for speech recognition." arXiv preprint arXiv:2005.08100 (2020).
> [2] Yu, Haonan, Sergey Edunov, Yuandong Tian, and Ari S. Morcos. "Playing the lottery with rewards and multiple languages: lottery tickets in rl and nlp." arXiv preprint arXiv:1906.02768 (2019).
> [3] Chen, Tianlong, Jonathan Frankle, Shiyu Chang, Sijia Liu, Yang Zhang, Zhangyang Wang, and Michael Carbin. "The lottery ticket hypothesis for pre-trained bert networks." arXiv preprint arXiv:2007.12223 (2020).

---

### Official Review · Reviewer_j7YG · 2021-11-11

**Correctness:** 4
**Technical Novelty And Significance:** 2
**Empirical Novelty And Significance:** 2
**Recommendation:** 6
**Confidence:** 3

**Main Review:**

Pros:
1. This paper is clearly written. The idea is simple and straightforward. Experiments are extensively performed, making the paper solid.

Cons:
1. The LTH basically follows the original LTH work and validates the claims from original LTH on ASR models. Though the experimental results are solid, the technical novelty of the paper is a bit thin.
2. The description of IMP could be elaborated. An algorithm or pseudo-code would make the pipeline more clear. Given that the overall framework does not change much, you should more focus on the ASR-specific details, e.g. how the pruning is different for 3 models? Besides, how does block sparsity work exactly?
3. In section 2, you mention pruning RNN-based models may be more challenging than pruning CNN-base models. But the method used in this paper is indeed similar to prior works. This claim is a bit inconsistent.
4. For the random tickets, do you randomly initialize a new $\theta_0'$ every time, or just use a $\theta_0'$ different from $\theta_0$?
5. As an empirical paper, it is important to give thorough experiments. For example, it's better for you to provide "existence" tables on CommonVoice and TED-LIUM as well, as Table 1.
6. "Table 4 shows qualititive results of ...". I think you mean "quantitative"?
7. It would be better if you can show some run time comparison tables.
8. Given that many modern ASR models now involves quantization (wav2vec2), would this affect how LTH works?

**Summary Of The Paper:**

In this paper, authors propose to use lottery ticket hypothesis (LTH) for ASR model pruning. The whole idea generally inherits from the original LTH paper. The model is first trained from scratch. Then you collect all the model weights and sort them. A proportion of the smallest weights are set and fixed to 0. Given this sparsified subnetwork, one further repeats the whole process and prune more parameters. After several rounds, if the pruned subnetwork still gives at least the same performance of the original full network , it is called a "winning ticket". The "winning ticket" subnetwork should be lightweight, transferrable, and noise-robust.
This work applies LTH to 3 models (CNN-RNN CTC, RNN-T, Conformer) on 3 datasets (TED-LIUM, Common Voice, LibriSpeech). The authors empirically show
1. The winning ticket exists for each method. The subnetwork could be only 4.4% size of the original full model, but still matches the performance.
2. Compared to other pruning methods, LTH performs best.
3. LTH with pretrained model performs better than LTH with random initialization.

Furthermore, the paper shows
1. One can enforce block sparsity for pruning to facilitate chip design.
2. The winning ticket can transfer to a new dataset and work well.
3. The pruned network is noise-robust.


**Summary Of The Review:**

The paper is well-written with a clear and simple idea inherited from prior works. LTH does perform well on ASR and this is a very nice finding. Though the paper claims pruning RNN-based models is different from pruning CNN in CV, in the end the method is still quite similar to the original LTH, which makes the paper's novelty a bit thin. But the work does have a pretty thorough experimental section including all kinds of verification tasks, and proves the effectiveness of LTH on ASR models. The paper would be better if you can give more details on how the smallest weights (or block sparsity) are pruned for different models.

---

> ### Author Response · Authors · 2021-11-12
> **Response to reviewer j7YG (part 1)**
>
> Thanks for your positive feedback on the solidarity, experiments, and organization, and we also appreciate your suggestions on overcoming the weaknesses. We provide a pointwise response to the Cons below.
>
> 1. Thanks for bringing this question up. As you may have noticed, a large number of LTH papers (many are high-visibility, e.g., [1][2]) are experimental-based, which focuses on verifying existing LTH properties of identifying new task-specific properties. Although these papers do not have novel methods, they do provide new insights and explanations to the existing phenomenon, which are also empirically novel and equally valuable to the community. In this similar vein, our paper verified the existence of LTH on different ASR models, and for the first time studied three new properties that are key to ASR applications, providing new insights and exciting results to both the model pruning and speech recognition community. This has been affirmed by reviewer etjM that “They conduct comprehensive analyses on several research questions and the verification on effectiveness of the two components (initial weight, and mask generated from IMP) in iterative weight magnitude pruning (IMP). They conduct advanced analyses on structured sparsity, transferability, and noise robustness, which echo the story told in the introduction part.”
> 2. Sorry for the confusion. We have added the pseudo-code of the algorithm to the revised manuscript at Section A.7.5 to make it easier to follow. Currently, we are using the same pruning method for the three ASR models (prune the 20% of remaining weights with the minimal magnitude at each iteration). In terms of block sparsity, it prunes a block of parameters (e.g., 1x4 block) instead of just one parameter, which has the minimal magnitude. Unstructured pruning could be thought as block sparsity with 1x1 block. We have also added these explanations to the revised manuscript at Section A.7.6.
> 3. RNNs usually have very different behaviors as CNNs or fully-connected networks, and we do observe that several pruning methods fail to generalize to RNN. As a result, experimental validations are required to answer the question of whether LTH can generalize to ASR (or RNN models). This is exactly one of the motivations of this study - conducting extensive experiments (Section 4) to empirically prove the existence of LTH in ASR (or RNN). In section 5, we go one step further, and explore three novel properties of LTH.
> 4. We randomly initialize a new $θ_0^′$ at each IMP iteration.
> 5. We have the results of CTC backbone on all three datasets in Table 2. We did not put other results in the original manuscript due to the computation resource and time limitation. We have actually run these experiments after the submission deadline, as shown in **Table S1** and **S2**. Winning tickets also exist in TEDLIUM and CommonVoice datasets. We have added these tables to the revised manuscript at Section A.7.1.
>
> **Table S1** Performance of the three backbones on TEDLIUM at the $\textit{extreme}$ sparsity or at the $\textit{best}$ performance. All the notations correspond to those in Table 1 of the manuscript.
>
> |backbone|$WER_{full}$|$WER_{ext}$|$RW_{ext}$|$WER_{best}$|$RW_{best}$|
> |:-:|:-:|:-:|:-:|:-:|:-:|
> |CTC|15.93|15.07|4.4\%|14.04|16.8\%|
> |RNN-Transducer|12.43|12.26|2.2%|13.96|41.0\%|
> |Conformer|7.33|7.20|1.8\%|6.98|21.0\%|
>
> **Table S2** Performance of the three backbones on CommonVoice at the $\textit{extreme}$ sparsity or at the $\textit{best}$ performance. All the notations correspond to those in Table 1 of the manuscript.
>
> |backbone|$WER_{full}$|$WER_{ext}$|$RW_{ext}$|$WER_{best}$|$RW_{best}$|
> |:-:|:-:|:-:|:-:|:-:|:-:|
> |CTC|5.57|5.41|16.8%|4.17|64.0%|
> |RNN-Transducer|3.41|3.39|10.7\%|3.02|26.2\%|
> |Conformer|1.35|1.35|8.6\%|1.16|32.8\%|

---

> ### Author Response · Authors · 2021-11-12
> **Response to reviewer j7YG (part 2)**
>
> 6. We will fix this in the revised manuscript.
> 7. We conduct the run time complexity evaluations and add the results to **Table S3, S4, and S5**. We use the code base at https://github.com/sovrasov/flops-counter.pytorch to compute MACs. The sequence length of the input spectrogram is computed from the average utterance lengths of the datasets. TED-LIUM: 5 seconds; Librispeech: 16 seconds; CommonVoice: 11 seconds. We have added these tables to the revised manuscript at Section A.7.2.
>
> **Table S3** Run time evaluation of the three backbones on LibriSpeech at the $\textit{extreme}$ sparsity or at the $\textit{best}$ performance. Here we use the Number of Multiply–Accumulate Operations (MACs) in Giga (G) to measure the run time complexity. We compute the percentage compared to full model for all the subnetworks. All the notations correspond to those in Table 1 of the manuscript.
>
> |backbone|$MACs_{full}$|$MACs_{ext}$|$MACs_{best}$|
> |:-:|:-:|:-:|:-:|
> |CTC|77.88G|20.1%|49.9%|
> |RNN-Transducer|124.56G|9.6%|39.8%|
> |Conformer|111.20G|7.7%|49.4%|
>
> **Table S4** Run time evaluation of the three backbones on TEDLIUM at the $\textit{extreme}$ sparsity or at the $\textit{best}$ performance. Here we use the Number of Multiply–Accumulate Operations (MACs) in Giga (G) to measure the run time complexity. We compute the percentage compared to full model for all the subnetworks. All the notations correspond to those in Table 1 of the manuscript.
>
>
> |backbone|$MACs_{full}$|$MACs_{ext}$|$MACs_{best}$|
> |:-:|:-:|:-:|:-:|
> |CTC|24.34G|16.0%|63.1%|
> |RNN-Transducer|38.92G|1.5%|39.9%|
> |Conformer|34.75G|1.3%|19.7%|
>
> **Table S5** Run time evaluation of the three backbones on CommonVoice at the $\textit{extreme}$ sparsity or at the $\textit{best}$ performance. Here we use the Number of Multiply–Accumulate Operations (MACs) in Giga (G) to measure the run time complexity. We compute the percentage compared to full model for all the subnetworks. All the notations correspond to those in Table 1 of the manuscript.
>
> |backbone|$MACs_{full}$|$MACs_{ext}$|$MACs_{best}$|
> |:-:|:-:|:-:|:-:|
> |CTC|53.54G|20.1%|49.9%|
> |RNN-Transducer|85.63G|9.2%|25.0%|
> |Conformer|76.45G|7.1%|31.2%|
>
> 8. Quantization reduces the capacity of each parameter of a model, and LTH and general model pruning reduce the number of parameters of a model. Both techniques will reduce the model capacity and expressivity. In my humble opinion, LTH would still hold with quantization, but the extreme sparsity will be lower. In addition, it also depends on the model complexity and training data size. As we observed in Section 4-RQ1, for a fixed dataset, the winning tickets extracted from larger models are sparser. For a fixed model, the winning tickets extracted from larger (more complex) dataset are less sparser. In the end, it is a dilemma between model size and data size.
>
>
> [1] Yu, Haonan, Sergey Edunov, Yuandong Tian, and Ari S. Morcos. "Playing the lottery with rewards and multiple languages: lottery tickets in rl and nlp." arXiv preprint arXiv:1906.02768 (2019).
>
> [2] Chen, Tianlong, Jonathan Frankle, Shiyu Chang, Sijia Liu, Yang Zhang, Zhangyang Wang, and Michael Carbin. "The lottery ticket hypothesis for pre-trained bert networks." arXiv preprint arXiv:2007.12223 (2020).

---

> ### Comment · Reviewer_j7YG · 2021-11-15
> **rebuttal for point 4 is missing**
>
> Could you further clarify point 4?

---

> > ### Author Response · Authors · 2021-11-15
> > **Thanks for your response**
> >
> > Thanks for your response and sorry for the confusion. We have actually answered point 4, but we put it at the last part of point 3 due to the formatting issue. We have edited the previous response to fix the problem. We repeat it here for your convenience.
> >
> > We randomly initialize a new $θ_0^′$ at each IMP iteration. Random ticket was suggested to be a competitive baseline by previous LTH papers (e.g., [1][2]).
> >
> > [1] Frankle, Jonathan, and Michael Carbin. "The lottery ticket hypothesis: Finding sparse, trainable neural networks." arXiv preprint arXiv:1803.03635 (2018).
> >
> > [2] Chen, Tianlong, Jonathan Frankle, Shiyu Chang, Sijia Liu, Yang Zhang, Zhangyang Wang, and Michael Carbin. "The lottery ticket hypothesis for pre-trained bert networks." arXiv preprint arXiv:2007.12223 (2020).

---

> > > ### Comment · Reviewer_j7YG · 2021-11-15
> > > **good clarifications**
> > >
> > > The clarifications are clear and the new results also addressed some of my concerns.
> > > In general, I think the strengths and weaknesses are clear for this paper. This paper is solid and experiments are thorough. The overall idea is to empirically validate the LTH performance on ASR models. The weakness is the novelty. Since this paper is basically validating the performance of an existing method on a new domain, the novelty is marginal. Given the new results and the updated version to address my concerns, I'm willing to update my score a bit.

---

> > > > ### Author Response · Authors · 2021-11-15
> > > > **Thanks for your positive reassessment**
> > > >
> > > > Thanks for your positive reassessment! We are glad to have a very good discussion with you. Please do let us know if there is anything that you believe we can do to improve it!

---

### Comment · Area_Chair_pRYB · 2021-11-15
**Please update your ratings if needed based on the authors' responses**

Dear Reviewers,

The authors have made detailed responses to all the reviews. Please take a look and see whether they address your concerns and update the ratings if necessary. Thanks for your help and expertise!

---

### Author Response · Authors · 2021-11-23
**General Response**

Dear AC and all reviewers:

We truly thank AC for encouraging discussions and really appreciate all the reviewers for their valuable suggestions. We are thankful for the reviewers now appreciating this work’s solid experiments and interesting conclusions.

- We thank reviewer **j7YG** for strongly acknowledging our solid and thorough experiments. We really appreciate reviewer **j7YG** for increasing our score.

- We thank reviewer **5kPn** for appreciating our efforts in the rebuttal and pointing out important references. Please kindly let us know if there is anything else we can address to convince you for upgrading the scores.

- We thank reviewer **5xPW** for the positive evaluation and acknowledging our interesting and useful conclusions.

- We sincerely hope to have further discussion with reviewer **etjM** to see if our response solves his/her concerns. We are confident that our response should have cleared the air, and we can clarify more if there is more need. We are happy to answer any additional questions and provide more information.

---

### Comment · Area_Chair_pRYB · 2021-11-24
**Reminder to update ratings if needed**

Dear Reviewers,

The authors have made detailed responses to all the reviews. Please take a look and see whether they address your concerns and update the ratings if necessary. Thanks for your help and expertise!

---

### Author Response · Authors · 2021-11-29
**Pre-decision: Summary of updates from Authors**

Dear AC and all reviewers:

We sincerely appreciate AC and all reviewers’ time and efforts in reviewing our paper. We truly thank all reviewers for their insightful and constructive suggestions, which help a lot in further improving our paper. We are thankful for the reviewers now appreciating this work’s solid experiments and interesting conclusions.


We have conducted several additional experiments and updates to this manuscript according to the reviewers comments and suggestions. Here is a summary of our updates:


* **[Improve the baseline performance to be closer to the reported SOTA]:** We have added RNN-LM, SpecAug, and comprehensive hyper-parameter tuning on the baseline models. The tuned baselines are much closer to the SOTA performances, and our results also suggest that we can still identify winning tickets that significantly surpass the full model performance under these new settings.
* **[Experiments with larger block size for block sparsity]:** Winning tickets can still be identified with larger block size (1x16 block).
* **[Experiments of adding noise (with different SNR) only at test time]:** As suggested by reviewer 5xPW, this is a more interesting setting, since this will demonstrate how robust the models are on out-of-distribution samples. Our results suggest that winning tickets are much more robust to out-of-distribution noise than dense models.
* **[Evaluations of the three backbones on other datasets]:** Winning tickets can be identified from all three backbones on all datasets.
* **[Run time evaluation of the three backbones]:** Run time complexities (MACs) corroborate with spatial complexities (sparsity), suggesting the spatial and temporal efficiency of the winning tickets.

We believe that these new results, together with our results reported in the original manuscript, should be comprehensive enough for this study. Meanwhile, we hope that our newly added experimental results are able to well address reviewer **ejtM**’s remaining concerns. Given the significant practical impact of this study to both model compression and ASR communities as recognized by reviewer **j7YG** and **5xPW**, we sincerely hope to have a potential discussion for a clear and fruitful resolution of the concerns raised during the discussion period. Please do not hesitate to reach out to us if there are other clarifications or experiments we can provide.

Thank you and best regards,

Authors

---

### Decision · Program_Chairs · 2022-01-20

**Decision:**

Accept (Poster)

**Comment:**

The paper presents a comprehensive analysis of lottery tickets hypothesis (LTH) on automatic speech recognition. The authors verified the existence of highly sparse “winning tickets” in ASR task, and analyzed its robustness to noise, transferable to other datasets, and supported with structured sparsity.

As agreed with the reviewers, the paper is well-written, the justification is thorough, and the finding that LTH does perform well on ASR is interesting. Though the novelty is marginal as it's a direct application of the LTH, this is the first investigation of LTH and brings new insights to the community.

The decision is mainly based on the thorough justification of the LTH to ASR and new insights it brings to the community.